# Few-femtosecond time-resolved study of the UV-induced dissociative dynamics of iodomethane

Lorenzo Colaizzi [1,2] ✉, Sergey Ryabchuk[2,3], Erik P. Månsson [1], Krishna Saraswathula[1], Vincent Wanie [1], Andrea Trabattoni [1,4], Jesús González-Vázquez [5,6] ✉, Fernando Martín [5,7] & Francesca Calegari [1,2,3] ✉

Ultraviolet (UV) light that penetrates our atmosphere initiates various photo-chemical and photobiological processes. However, the absence of extremely short UV pulses has so far hindered our ability to fully capture the mechanisms at the very early stages of such processes. This is important because the concerted motion of electrons and nuclei in the first few femtoseconds often determines molecular reactivity. Here we investigate the dissociative dynamics of iodomethane following UV photoexcitation, utilizing mass spectrometry with a 5 fs time resolution. The short duration of the UV pump pulse (4.2 fs) allows the ultrafast dynamics to be investigated in the absence of any external field, from well before any significant vibrational displacement occurs until dissociation has taken place. The experimental results combined with semi-classical trajectory calculations provide the identification of the main dissociation channels and indirectly reveal the signature of a conical intersection in the time-dependent yield of the iodine ion. Furthermore, we demonstrate that the UV-induced breakage of the C-I bond can be prevented when the molecule is ionized by the probe pulse within 5 fs after the UV excitation, showcasing an ultrafast stabilization scheme against dissociation.

It is well known that the ultraviolet (UV) radiation penetrating our atmosphere triggers a large number of photochemical and photo-biological processes[1,2]. One of the fundamental mechanisms governing these photointeractions is the relaxation of the biomolecular building blocks from the photoexcited state back to the ground state via non-adiabatic crossings, also known as Conical Intersections (CIs). These rapid relaxation paths can take place already within a few femtoseconds[3], thus requiring ultrafast laser spectroscopy methods for time-resolved studies in the laboratory.

Iodomethane (methyl-iodide, $CH_3I$) is a well-known molecule in which a single UV-photon transition leads to a rapid cleavage of the C–I bond. It has been frequently employed since the seminal work of Zewail et al.[4] for benchmarking theoretical predictions[5–7]. The first absorption band of iodomethane (A-band) is characterized by a broad absorption continuum from 230 to 300 nm (4.1–5.4 eV) with a maximum near 260 nm (4.8 eV)[6,8] (Supplementary Fig. 1a). Figure 1 shows the calculated potential energy surfaces highlighting a few relevant states for both the neutral molecule and the cation. UV

[1]Center for Free-Electron Laser Science, Deutsches Elektronen-Synchrotron DESY, Hamburg, Germany. [2]Physics Department, Universität Hamburg, Hamburg, Germany. [3]The Hamburg Centre for Ultrafast Imaging, Universität Hamburg, Hamburg, Germany. [4]Institute of Quantum Optics, Leibniz Universität Hannover, Hannover, Germany. [5]Departamento de Química, Universidad Autonoma de Madrid, Madrid, Spain. [6]IADCHEM, Universidad Autónoma de Madrid, Madrid, Spain. [7]Instituto Madrileño de Estudios Avanzados en Nanociencia (IMDEA-Nanoscience), Madrid, Spain. ✉e-mail: lorenzo.colaizzi@polimi.it; jesus.gonzalezv@uam.es; francesca.calegari@desy.de; francesca.calegari@uni-hamburg.de

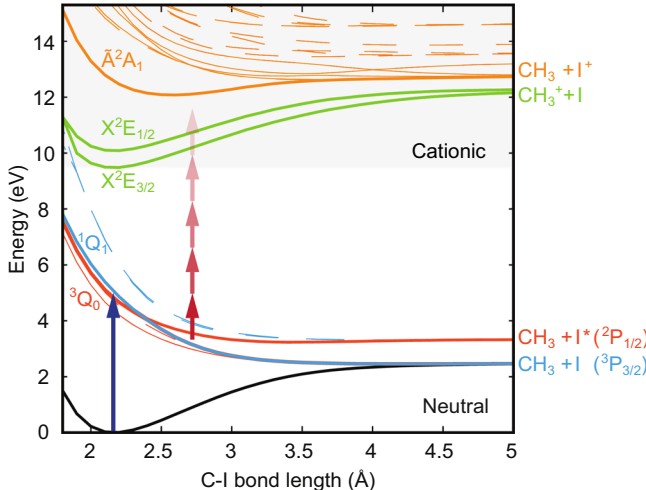

**Fig. 1 | Potential energy curves of iodomethane along the C-I bond.** One-dimensional potential energy curves were calculated for iodomethane as a function of the C-I bond length, for both the lowest neutral and cationic electronic states. The states responsible for the observed excited state dynamics are highlighted with solid lines, along with their dissociation products. Dashed lines are used to indicate states included in the calculation, which negligibly contributes to the observed dynamics. A single UV photon (blue arrow) excites the molecule from its ground state to the neutral $^1Q_1$ and $^3Q_0$ states (blue and red, respectively), through which it dissociates. Multiphoton absorption by the probe (red arrows) is used to follow the neutral dynamics by reaching different cationic states that lead to the production of charged fragments.

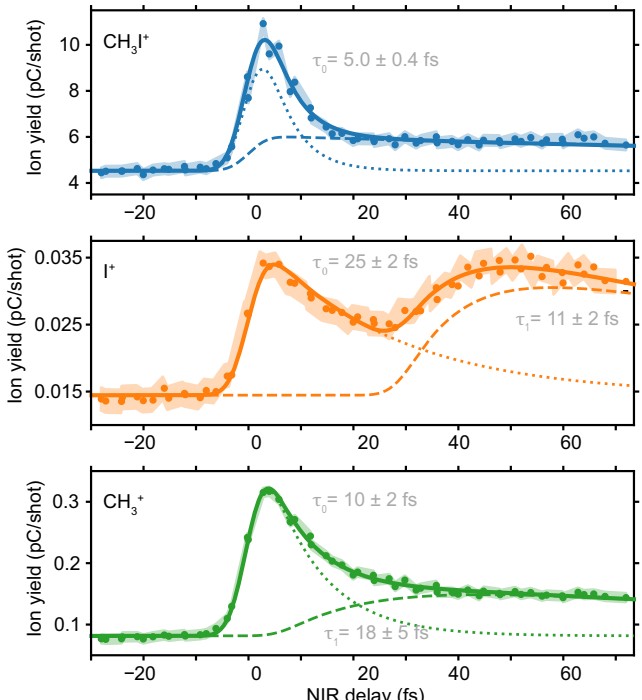

**Fig. 2 | Time-resolved experimental results.** Experimental data (dots) for each fragment detected as a function of the pump-probe delay (positive delay means the NIR pulse interacting with the molecule after the UV pulse) and the curve fitting model (solid lines) described in the text. The short-lived (dotted) and long-lived (dashed) contributions to the curve-fitting model are also reported. The shaded area represents the delay-averaged standard error obtained from two measurements (Supplementary Note 3). The ion yield is reported as an amplified charge in pC (recorded by the acquisition system) per laser shot (details provided in the "Methods" section).

photoabsorption brings 98% of the excited population to the dissociative $^3Q_0$ state, and the rapid intersection with the $^1Q_1$ state leads to two possible reaction pathways that differ only by the electronic excitation of the produced iodine atom.

The presence of the conical intersection near the Franck–Condon (FC) region[9–12] makes iodomethane a popular test case for investigating the effects of UV-induced non-adiabatic crossings at ultrafast timescales. Time-resolved experiments have so far been performed with UV pulses longer than the time required to reach the conical intersection, respectively reporting instrumental response functions (IRF) of 118 fs[13], 60 fs[11], 26 fs[14], and 24 fs[10] full-width at half maximum (FWHM) with very different methods from extreme ultraviolet (XUV) transient absorption to infrared Coulomb explosion imaging. Despite this limitation, the work of Chang et al.[10] (shortest pump pulse to date) could extract a value of 15 ± 4 fs to reach the conical intersection. Still, substantial uncertainty remains in the time window between 5 and 30 fs, where the non-adiabatic effects dominate the ultrafast dissociative dynamics. Nevertheless, shorter UV pulses are required to induce a sudden electronic excitation before any significant structural rearrangement can take place. So far, this scheme remains unexplored due to the challenges in producing and utilizing few-femtosecond UV pulses for time-resolved measurements, which have been surmounted only very recently[15–19].

In this work, we report on the combined experimental and theoretical investigation of the ultrafast dissociation dynamics of the benchmark molecule iodomethane following the sudden excitation by an ultrashort UV pulse. In particular, we use a (4.2 ± 0.3) fs UV pump pulse, corresponding to <7 % of the dissociative C–I vibrational mode period, to ensure that the molecule is nearly frozen during the excitation step.

## Results

In our experiment, an ultrabroadband UV pulse, with a duration of 4.2 ± 0.3 fs, photoexcites a manifold of dissociative Q states. The spectrum covers most of the absorption band of $CH_3I$ (Supplementary

Fig. 1). After excitation, a delayed near-infrared (NIR) pulse, with a duration of 5.6 ± 0.2 fs, probes the UV-induced dynamics through multi-photon ionization. The produced ions are detected in a time-of-flight mass spectrometer as a function of the pump-probe delay (see Methods). Only three ions are produced: the parent ion $CH_3I^+$ of molecular weight 142 μ and the two fragments iodine $I^+$ (127 μ) and methyl $CH_3^+$ (15 μ). Figure 2 reports their delay-dependent yield.

All fragments show a clear maximum near the zero-time delay, while different dynamics can be observed at longer delays. The parent ion $CH_3I^+$ is the most abundant ion produced in our experiment since its maximum yield is almost two orders of magnitude higher than the one of the methyl fragment and three orders of magnitude higher than the one of the iodine fragment. To better quantify the yield variations for each ion, we use a curve-fitting model composed of three terms: a rapid exponentially decaying peak, a delayed long-lived contribution, and a baseline. The model also includes the convolution with a Gaussian term representing the pump-probe cross-correlation. Details are provided in Supplementary Note 4. The fitting curves are reported in dotted lines for the short-lived contribution and in dashed lines for the long-lived contribution in Fig. 2, and the parameters are summarized in Supplementary Table 1.

The $CH_3I^+$ signal decays very rapidly with a time constant $\tau_0 = 5.0 \pm 0.4$ fs. It also exhibits a weak long-lived contribution that does not decay within our observation window. Since absorption of a single UV photon only leads to dissociative states, we conclude that this long-lived contribution is due to a two-UV-photon excitation of Rydberg states just below the ionization threshold[20]. Electrons promoted to the long-lived Rydberg states can be ionized by one NIR photon nearly independently of the positive time delay, leading to an

almost constant contribution. A similar long-lived contribution has been reported in previous works for one XUV-photon absorption[21] and two-UV-photon absorption[10,14], the latter with similar pump intensities.

Unlike the intact iodomethane ion, the iodine fragment signal exhibits a fast decay $\tau_0 = 25 \pm 2$ fs followed by a remarkable increase of around 28 fs delay. Similar behavior is observed in the delay-dependent yield of the methyl fragment: the signal first decays rapidly with a time constant $\tau_0 = 10 \pm 2$ fs and then more slowly reflecting a long-lived contribution. The presence of fragment ions, even at zero-time delay, when the neutral dissociation has not yet occurred, shows that the NIR probe pulse promotes the system to highly excited states of the cation that are known to be dissociative. In particular, in Fig. 1, we highlight the excited cationic state Ã$^2$A$_1$, which results in the production of the CH$_3$ and I$^+$ fragments due to its shallow potential well. We can already mention that our calculations indicate that the fragment ions are produced via NIR-induced multiphoton transition to this state as well as to the X$^2$E$_{3/2}$ and X$^2$E$_{1/2}$ states.

To further substantiate the interpretation of the measured delay-dependent yield, we performed full-dimensional classical trajectory calculations using a modified version of the surface hopping approach including the arbitrary couplings (SHARC) method[22,23], which combines XMS-CASPT2 spin-free energies and gradients with Perturbed Modified CASSCF spin-orbit terms. Additional details about the computational methods are provided in the Methods section. This methodology was also used to evaluate the potential energy curves. In this approach, the gradients of the spin-orbit states are obtained as linear combinations of the spin-free gradients. The nuclear derivative of the spin-orbit Hamiltonian (very localized around the heavy atom) is neglected. With this method, a large set of trajectories is launched from the dissociative states that are accessible by the absorption of one UV photon from the ground state. In order to simulate the broadband pulse, the trajectories are weighted using the experimentally measured UV spectrum, assuming that the transition dipole moment values are almost identical within this energy range[24]. Since ab initio calculations, including the multiphoton NIR probing step, are not computationally feasible due to the rather large number of NIR photons needed to ionize the molecule, we have adopted the following simplified approach. First, we compute the energy differences between the excited states of the neutral molecule and the states of the molecular cation, as a function of time along every launched trajectory. In this way, we can dynamically quantify the number of NIR photons required to ionize the molecule as the nuclei propagate along the potential energy curves of the neutral states. Subsequently, for each trajectory, we compute the total energy $E_{kin} + E_{pot}$ acquired by the nuclear trajectories. At each time, the trajectory is considered to contribute to a specific fragment ion $i$ when

$$E_{kin} + E_{pot} \geq D_i \qquad (1)$$

where $D_i$ is the minimum energy required for dissociation along the potential energy curve leading to fragment $i$. We use Eq. (1) to assign each trajectory to the intact parent (CH$_3$I$^+$) ion, the methyl fragment ion (CH$_3^+$), or the iodine fragment ion (I$^+$). Finally, at each time step, we associate a trajectory leading to an ion $i$ with a specific number of absorbed NIR photons, $n$, when such trajectory lies in the energy range between $n-1$ and $n$ NIR photons. Then, we assume that the probability of producing an ion $i$ at a given time after the absorption of $n$ photons is simply given by the total count of trajectories leading to the ion $i$ in the corresponding energy region. In doing so, we assume that the ionization cross sections are independent of the photoelectron energy and they are identical for all final states of the cation. Figure 3 (a–c) shows the number of trajectories (here represented with different colors) leading to the different ionic species as a function of time and ionization energy. We note that our theoretical approach cannot model resonance-enhanced multi-photon ionization (REMPI).

Although detailed knowledge about the resonances is, in general, needed for a proper interpretation of transient signals of molecular dynamics[25], previous works on iodomethane[26,27] indicate that REMPI does not play a relevant role in our experimental conditions.

Using this simple approach, we predict that there are many more trajectories leading to I$^+$ than to the other ions. We can also assign many of the features observed experimentally in the delay-dependent cationic yields. In the case of the CH$_3$I$^+$ ion (see Fig. 3a), we can see that, immediately after the sudden UV-excitation, a minimum number of three NIR photons is required to reach the X$^2$E$_{3/2}$ ground state of the cation. Due to the steepness of the potentials of the involved excited states of the neutral molecule, access to a stable and intact CH$_3$I$^+$ ion via three-NIR-photon ionization is only possible within the FC region. Soon after, the energy separation increases, and more NIR photons are required to ionize the neutral molecule. As Fig. 3a shows, in the first 20 fs, only pathways involving 3 or 4 NIR photons can produce CH$_3$I$^+$. The ground state of the cation could also be accessed at later times via a five-photon transition, however at a bond length of 3 Å, the molecule has gained more than 1 eV of nuclear kinetic energy, and this excess of energy leads to a fast dissociation even when the molecule is promoted to the ground state of the cation. As a result, the rapid decay of the number of trajectories leading to CH$_3$I$^+$ (Fig. 3a) is complemented by the sudden increase of trajectories that dissociate into CH$_3^+$ (Fig. 3c). Furthermore, this rapid disappearance of a stable CH$_3$I$^+$ agrees well with the experimentally observed 5-fs decay of the CH$_3$I$^+$ yield.

In contrast to the mechanism discussed above, the production of iodine ions requires the absorption of more NIR photons (see Fig. 3b). As the probability of multiphoton NIR absorption drops with the number of photons (not reflected in these colormaps), the lowest potential energy curves of the cation are expected to be the relevant ones to interpret the experimental results. The five-photon transition to the Ã$^2$A$_1$ state is present at early times and decreases within 20 fs. Subsequently, six- and seven-NIR-photon transitions are necessary since the energy separation between the neutral and cationic states rapidly increases (see Fig. 1). The experimentally observed delayed increase of the I$^+$ yield can be then attributed to the six- or seven-photon transitions. After 30 fs, the ionization energies remain nearly constant over time (see Fig. 3b) as a result of the potential energy curves of the cationic states being almost parallel to those of the neutral states along the reaction coordinate. Finally, we mentioned above that the methyl fragment (Fig. 3c) mostly originates from the ground state of the cation when the transition occurs long after the FC region. For this reason, there are no trajectories contributing to the formation of CH$_3^+$ around time zero. However, an enhancement of the methyl ion signal is observed experimentally around zero-time delay. This can be assigned to an internal conversion mechanism (e.g., via the unbound continuum of the lower-lying X$^2$E) that allows part of the population promoted to the Ã$^2$A$_1$ state via a five-photon transition to relax to the ground state of the cation[21,28]. Similarly, from higher-lying states of the cation, prompt dissociation to I$^+$ is favored, but internal conversion from high-lying B̃ (not shown) to Ã and then to X states for the production of CH$_3^+$ has also been reported with branching ratios dropping with increasing photon energy[28]. Our experimental results show a reduced ratio of CH$_3^+$ to I$^+$ after the initial short-lived peak, which could indicate that the conversion from higher to lower potential energy curves is less likely when the NIR photoionization occurs after the C–I bond elongation. The above-described internal conversion mechanism is not fully included in the present time-dependent theory. Therefore, no trajectories appear before 20 fs (see Fig. 3c) in contrast with the clear maximum observed at the zero-time delay for the measured CH$_3^+$ yield.

## Discussion

To obtain a visual comparison with the experimental data, the 2D maps have been energy-integrated within a selection per number of NIR

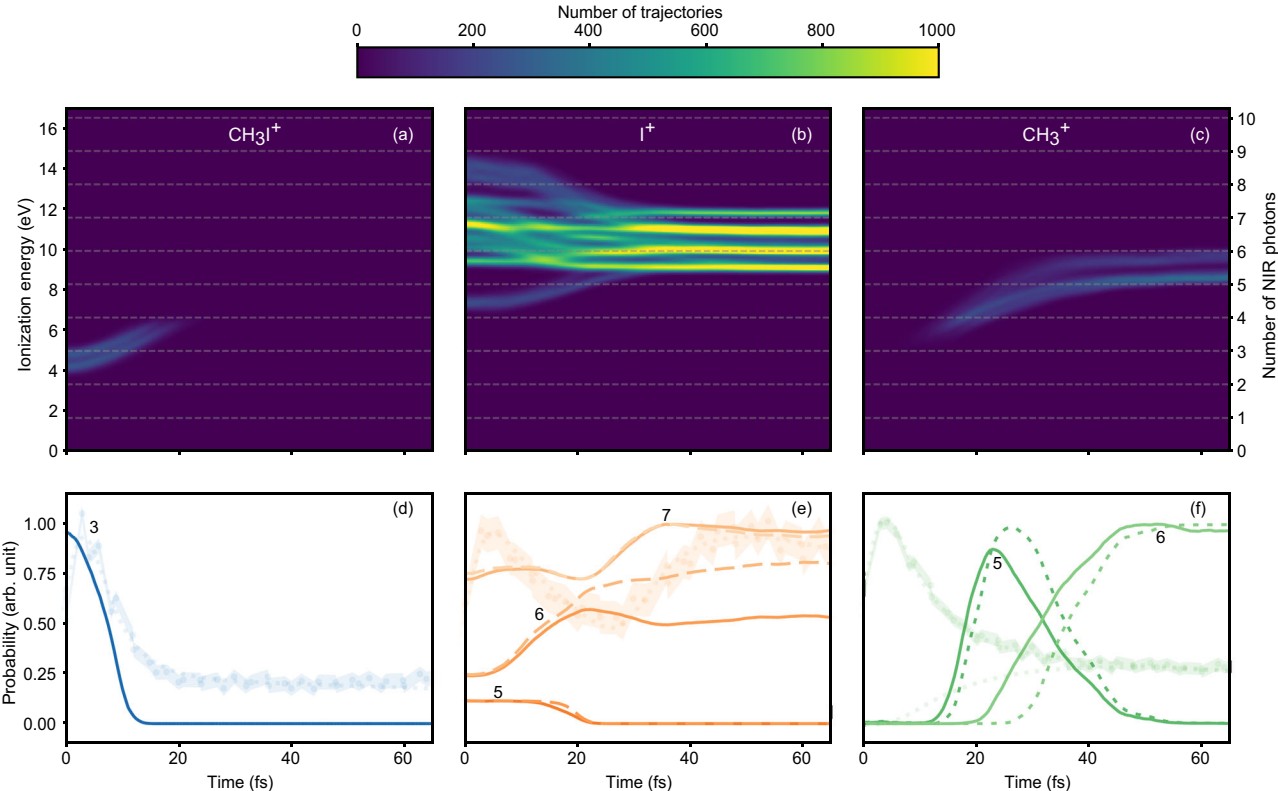

**Fig. 3 | Trajectories obtained by semiclassical calculations in neutral CH₃I.** The trajectories are classified by the kinetic energy release in case of ionization to various final states: (**a**) trajectories that remain bound as CH₃I⁺, (**b**) trajectories with sufficient energy to produce I⁺, and (**c**) trajectories that lead to CH₃⁺. The colormap shows the number of trajectories contributing to the creation of the respective ions without considering ionization probabilities. Horizontal dashed lines indicate multiples of the central NIR photon energy (1.65 eV) and allow us to identify the number $n$ of NIR photons required to reach the ionization threshold. Bottom panels (**d–f**): Solid lines show the number of trajectories leading to fragment $i$ in the ($n$-1,$n$) energy region (see main text). The dashed line in (**e**, **f**) represents the results obtained by considering only trajectories ending in the I*(²P₁/₂) channel (i.e., no transfer of population from the ³Q₀ to the ¹Q₁ state). The numbers indicate the number of NIR photons that must be absorbed in each case. For comparison, the experimental data are shown as a shaded area in the background, together with some of the individual contributions obtained from the fit (dotted lines), each normalized to a maximum of 1.

probe photons. The main results are displayed in Fig. 3(d–f) for each ion (more details in Supplementary Note 5). The experimental data and selected fitted curves are also reported as dotted and full lines, respectively, for comparison. We must stress that, within this theoretical framework, the individual multiphoton contributions are not weighted by a cross-section and, therefore, we cannot aim at a quantitative comparison with the experimental data. However, this comparison allows the main features observed in the time-resolved measurement to be assigned to specific multiphoton channels. For the CH₃I⁺ parent ion (Fig. 3d), the dominant contribution comes from the three-photon channel, which decays very similarly to the experimental data. A very minor contribution can also be expected from the four-photon channel (Supplementary Fig. 6). As discussed above, as the trajectories move away from the FC region, the transition to the cationic ground state requires additional NIR photons, and ultimately the cation will dissociate into CH₃⁺ + I. The experimentally observed delayed creation of CH₃⁺ is primarily assigned to the five-photon and six-photon channels (see Fig. 3f). In the case of I⁺, a large number of channels is possible, and the assignment is not trivial. However, the comparison between theory and experiment clearly indicates that, in this case, primarily the five-, and seven-photon channels are responsible for the observed dynamics, leading to a depletion of the signal at around 25 fs as in the experiment (see Fig. 3e). Absorption of 7 NIR photons at the longer times results in the dominant process because (i) the probe-free I*/(I + I*) and I/(I + I*) branching ratios are 0.68 and 0.32, respectively, so that I⁺ can be produced by ionizing both I and I*, and (ii) the total number of accessible I⁺ states from I and I* is

more than three times larger for seven-photon than for six-photon absorption.

While direct comparison between theoretical curves and experimental data is not straightforward, these curves still offer valuable insights. We used a best-fit approach to estimate the contribution of each theoretical curve to the experimental signal for every ion species by assuming that the experimental signal is represented by a linear combination of these curves. To this goal, all the curves have been convoluted with the IRF to include the resolution given by the finite duration of our pulses. Detailed results can be found in Supplementary Note 5. For the parent ion, the only coefficient that is significantly different from zero is that of the three-photon curve. For the methyl ion, a combination of only five- and six-photon curves best fits the experimental data. The most interesting case is the one of the iodine ion, for which the largest contributions come from the five- and seven-photon curves. Indeed, as can be seen in Fig. 3e, the shape of the six-photon curve cannot reproduce the experimental signal.

Assuming that the absorption probability is higher for a low number of photons, these assignments are consistent with the corresponding ion yields observed in our experiments. CH₃I⁺ is the most abundant ion produced because it only requires a three-photon transition. The parent ion is then followed by CH₃⁺, which requires a five-photon absorption, while I⁺ is the least produced ion because it primarily requires the absorption of seven photons.

We show next that the signature of the conical intersection between the ³Q₀ and ¹Q₁ states is imprinted in the time-evolution of the measured ion yields. For this, we have performed calculations like

those shown in Fig. 3, but now taking only into account trajectories ending into the $I(^2P_{1/2})$ channel, i.e., no transfer of population from the $^3Q_0$ to the $^1Q_1$ state. The results of these calculations are shown by dashed lines in Fig. 3e–f (and Supplementary Fig. 7). As can be seen, there are no significant changes in the dynamics of the $CH_3I^+$ and $CH_3^+$ yields. However, for the $I^+$ yield, the number of trajectories requiring the absorption of six photons is now comparable to that requiring the absorption of seven photons, in contrast with the results of the full calculation in which the number of trajectories requiring the absorption of 7 photons clearly dominates. As six-photon absorption is more likely than seven-photon absorption, the measured yields should mainly follow the six-photon curve in this artificial scenario. However, this is not the case: the six-photon curve does not exhibit the characteristic minimum observed in the measured yield at around 20–30 fs (a decay followed by an increase of the $I^+$ yield). This behavior is, however, very well described by the seven-photon curve. As Fig. 3e shows, when the passage through the conical intersection is correctly described, there are many more trajectories requiring the absorption of seven photons than six photons (solid lines in the Figure), better matching the non-monotonous behavior observed in the experiment. We can thus assign this behavior to the presence of the conical intersection.

To verify if the multiphoton ionization process associated with the probe step may preclude the observation of the pump-induced dynamics in the experiment, we have performed additional calculations based on the solution of the Time-Dependent Schrödinger Equation (TDSE). In these calculations, the NIR-induced dipole couplings between the bound electronic states of the neutral molecule and between the electronic states of the molecular cation, respectively, are explicitly included for all states shown in Fig. 1. The couplings between those bound states and the electronic continuum states, which are associated with the various dissociative channels of the molecular cation, are represented by the corresponding Dyson norms. Here, the Dyson norms are conveniently rescaled to account for the multiphoton nature of the transition (more details in Supplementary Note 5). As in the case reported in Fig. 3d, the $CH_3I^+$ yield resulting from this model decays very rapidly and follows the experimental trend, and the $I^+$ yield exhibits a non-monotonous dependence as a function of time as in the case reported in Fig. 3e. The absence of any significant difference in the extracted time-dependent probabilities in between the two models indicates the robustness of the discussed features and supports the validity of our assignments.

Theoretical calculations also allow us to extract the time required to reach the CI between the neutral states $^3Q_0$ and $^1Q_1$. Since the pump pulse spectrum used in this work covers nearly the full UV absorption band (Supplementary Fig. 1a), we can evaluate this time as a function of the excitation energy.

Figure 4 reports the time at which the $^3Q_0$ and $^1Q_1$ states become degenerate for every launched trajectory. From this figure, a clear correlation can be observed between the excitation energy of the UV-pump pulse and the extracted time to reach the CI. The dotted red line represents the best linear fitting, and it results in a slope of $10.47 \pm 0.76$ fs/eV, obtained from a Deming (total least squares) regression[29]. The positive slope indicates that a trajectory launched at higher potential energy, i.e., shorter initial bond length in Fig. 1, requires more time to reach the CI. For our UV spectrum (bottom panel of Fig. 4), centered at 4.9 eV, the average time to reach the CI is 9.9 fs (left panel of Fig. 4), and the sample standard deviation among 374 trajectories is 2.4 fs. A previous investigation based on single photon excitation of 4.48 eV[10] and 4.63 eV[13] resulted in an estimated time to reach the CI of approximately 13 fs, a larger value compared to the theoretical prediction reported here. We assign the discrepancy to different theoretical modeling used in the above-mentioned works, which is based on the extended multistate variation of the CASPT2 theory[30].

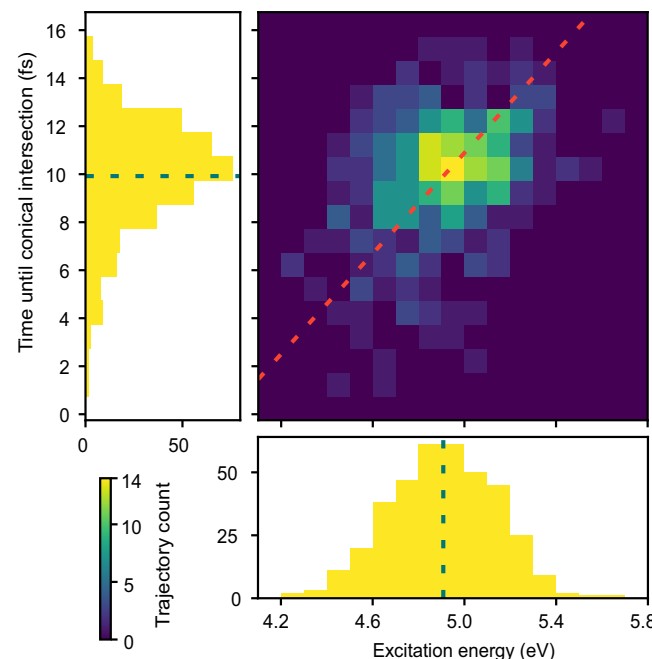

**Fig. 4 | Time required to reach the conical intersection between the $^3Q_0$ and $^1Q_1$ states as a function of the pump excitation energy.** Trajectories are sorted by their initial UV-excitation energy (horizontal axis) and their arrival time (vertical axis). The dotted red line is fitted with a slope of $10.47 \pm 0.76$ fs/eV for the energy dependence of the arrival times. Dotted lines in the 1D histograms indicate the mean values.

Finally, we further elaborate on the measured and theoretically confirmed decay of the parent ion signal within 5 fs. Our experiment clearly indicates that, if ionization by the probe pulse takes place after 5 fs, the corresponding cation inevitably breaks. However, this can be avoided if ionization by the probe occurs within a very short time window. This very short time window is a combined result of the steepness of the dissociative neutral curves ($^3Q_0$ and $^1Q_1$) and of the deep and narrow potential energy curve of the ground state of the cation ($X^2E_{3/2}$ in Fig. 1).

In Fig. 5, we graphically summarize the stabilization mechanism: the absorption of one UV photon promotes the molecule to highly dissociative curves, which very quickly leads to the cleavage of the C–I bond (Fig. 5a). This dissociation pathway can be prevented by immediately ionizing the molecule with a NIR pulse before the bond elongates further (Fig. 5b), i.e., within the first 5 fs. To emphasize the importance of employing few-femtosecond pulses in both the pumping and the probing steps, we re-evaluated our fitted curve model with the 24-fs FWHM instrumental limit of a previous study[10]. The use of such longer pulses would reduce the stabilizing effect to about 40–50% of what is obtained in the present work. An even lower percentage (13–26%) would be obtained with an 80 fs instrumental limit[31].

In summary, we have investigated the UV-induced ultrafast photodissociation dynamics of $CH_3I$ with unprecedented time resolution. Multiphoton ionization by a few-optical-cycle NIR probe pulse is used to follow the nuclear dynamics of the neutral molecule. The comparison between the measured delay-dependent ion yields and a theory model based on semi-classical trajectories simulation allows both potential energy surfaces of the neutral molecule ($^3Q_0$ and $^1Q_1$) as well as several cationic states involved in the dissociative dynamics to be mapped during the first 60 fs after the prompt UV excitation. Despite the many channels opened by the multiphoton NIR probing, the use of a 4.2 fs UV pulse allows the theoretical modeling to be precisely benchmarked since the nuclei propagate freely after the ultrashort UV pulse ends. From the comparison between experiment and theory, we

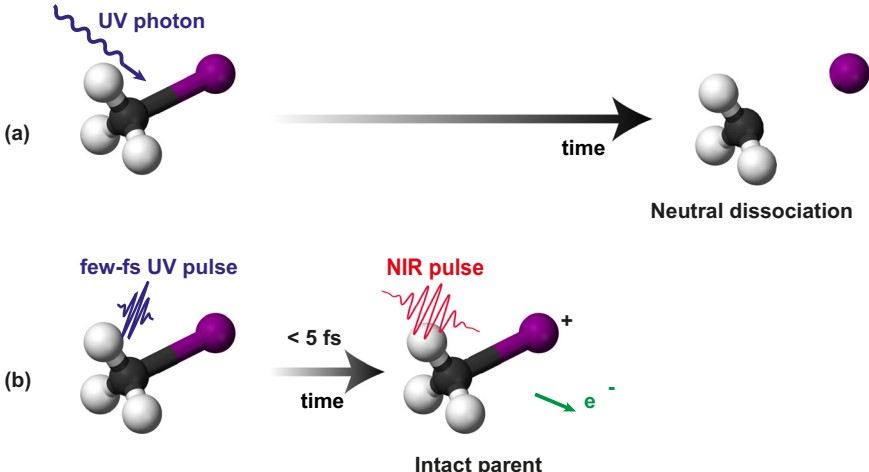

**Fig. 5 | Production of the intact parent ion. a** Exposure to UV radiation inevitably leads to a rapid neutral dissociation. **b** Prompt NIR ionization within a short time window of 5 fs can be used to prevent molecular fragmentation.

could identify indirect signatures of the presence of the conical intersection in the measured time-dependent yield. Our theoretical model is further used to disclose the time it takes to reach the conical intersection between the neutral states as a function of the excitation energy. Never observed in previous iodomethane studies, we demonstrate a stabilization pathway against the rapid UV-induced dissociation of $CH_3I$ using a sequence of few-optical-cycle UV and NIR pulses. The stabilization probability decays with a lifetime of only 5 fs. Our work demonstrates the benefits of using ultrashort UV pulses to better examine reaction pathways that are relevant for UV photo-chemistry and offers new important control perspectives for photo-reactions. Looking forward, the combination of such ultrashort UV pulses with attosecond extreme ultraviolet or soft-x ray pulses will soon give access to electronic mechanisms such as charge migration in neutral molecules triggered by broadband UV radiation and their temporal evolution in the presence of non-adiabatic crossings.

## Methods
### Experimental setup
The pump-probe experiment was performed with a multipass Ti:sap-phire amplifier (Femtopower, Spectra-Physics), producing 25 fs CEP-stabilized laser pulses with a pulse energy of up to 10 mJ at a 1 kHz repetition rate. Compression to 5.6 fs pulses is achieved by coupling a portion of the laser beam (6 mJ) into a hollow-core fiber setup filled with a pressure gradient up to 1.7 bar of helium and combined with multiple reflections on a pair of chirped mirrors (Few-cycle Inc.)[32,33]. The produced few-optical-cycle NIR pulses are temporally character-ized using an SH-FROG (Second Harmonic – Frequency Resolved Optical Gating)[34] set-up placed at the entrance of the beamline[35]. Inside the beamline, the NIR beam is split into two replicas: one serves as the probe pulse (80 μJ), while the other (450 μJ) is focused in a glass cell filled with neon at high pressure to produce the UV pump pulse via third harmonic generation (THG)[16,18]. While gaseous media have low conversion efficiencies that require high pressures, they are key to preserving the temporal characteristics of the UV, as dispersion effects are minimized[17] The UV pulse is separated from the generating NIR pulse by using the reflection from silicon mirrors at Brewster's angle. The resulting UV pulses have a broadband spectrum covering the 4.4–5.4 eV energy range (Supplementary Fig. 1a), which corre-sponds to a transform-limit duration of 2.7 fs. The peak intensity of the UV pump pulse can be controlled by tuning the gas pressure in the THG cell. For this work, the UV energy per pulse was set to 90 nJ, corresponding to an estimated intensity on target of $3.8 \times 10^{12}$ W/cm$^2$. The UV pulse duration has been estimated in situ through two-color

cross-correlation in Krypton and resulted in FWHM duration $W_{UV} = 4.2 \pm 0.3$ fs (Supplementary Note 1).

The NIR probe's central wavelength corresponds to 1.65 eV, with a bandwidth of 0.6 eV, as shown in Supplementary Fig. 1b. The NIR pulses are focused on the target by a gold-coated toroidal mirror with the peak intensity adjustable over a range from $9.2 \times 10^{11}$ W/cm$^2$ to $1.2 \times 10^{13}$ W/cm$^2$ using a motorized iris. The lowest intensity was chosen for the time-resolved measurements to minimize the background of the probe-only signal. In this condition, the parent ion background pro-duced by the NIR-only pulse is $5.5 \times 10^{-4}$ times lower than the corre-sponding two-color signal (Supplementary Note 2). This relatively low intensity also ensures that strong-field effects can be minimized while still observing multiphoton ionization. The UV pulses are focused on the target by a spherical mirror (90-cm focal length) and recombined with the probe in a non-collinear geometry with an angle of 0.6°. From the characterized FWHM durations of the UV and NIR pulses and the number of absorbed NIR photons in the probing step, the experimental time resolution is estimated to be $5.1 \pm 0.3$ fs.

$CH_3I$ is delivered to the interaction chamber in the gas phase from a 3 mL stainless steel container filled with liquid iodomethane (Sigma-Aldrich). The molecules are then injected into the interaction region via a 0.5-mm diameter tube nozzle and a 2-mm skimmer to keep the pressure in the interaction chamber at $2.2 \times 10^{-6}$ mbar. A reflectron time-of-flight mass spectrometer[36] detects the product ions via amplification in a triple-layer (Z-stack) microchannel plate, whose output pulse is recorded by an analog-to-digital converter. The recorded voltage is divided by the 50 Ω characteristic impedance to obtain a current. The current's integral over the short time interval corresponding to a specific ion in the time-of-flight mass spectrum gives the detected charge. In Fig. 2, we show the average charge per laser shot in picocoulombs with the units expressed as pC/shot.

### Theory
The electronic structure for iodomethane was calculated using the XMS-CASPT2 method with an imaginary shift of 0.3 Hartree[30]. Four active orbitals were considered in the preceding CASSCF calculation ($\sigma_{CI}$-bonding, $\sigma_{CI}^*$-antibonding, and two lone pairs of the iodine atom), giving a CAS(6,4) for the neutral and a CAS(5,4) for the cation. Four different sets of orbitals were optimized by state-averaging different numbers of roots. For the neutral, 3 singlets and 3 triplets were considered, whereas for the cation, 11 doublets and 3 triplets were necessary to describe the dissociation limit. In order to include relativistic effects, we combine an ANO-RCC basis

set[37,38] contracted to triple zeta valence polarized (ANO-RCC-VTZP) with a second order Douglas-Kroll Hamiltonian[39] and an AMFI approximation[40] for the spin-orbit coupling. This protocol was previously compared with experimental data in references[7,21] for neutral and cation, respectively.

An interface between the BAGEL[41] and OpenMolcas[42,43] codes was implemented for the calculations. On the one hand, BAGEL was in charge of the orbital optimization and the XMS-CASPT2 calculation (including analytical gradient)[30]. On the other hand, spin-orbit and overlap calculations were done using the OpenMolcas code, both at perturbation-modified CASSCF (PM-CASSCF). The passing of the wave function between both codes was performed by transforming the PM-CASSCF CI-vector between determinants (BAGEL) and CSFs (OpenMOLCAS).

Finally, for the dynamics, the previous protocol was connected with a revised version of the SHARC[22,23] method. In contrast with the original paper, the propagation of the electronic wave function was done using diabatic states, and surface hopping probability was evaluated using the flux formula[44]. The use of the SHARC method to describe the photodissociation of iodomethane has already been tested and compared with experimental data in previous works[7,13,20,45]. Decoherence is done using the Energy Decoherence method[46] with a 0.1 Hartree coefficient. Since we are not calculating the non-adiabatic vector, velocity scaling after hop is directly done in the direction of the trajectory. The nuclear time step was 0.5 fs, and a total of 50 substeps were interpolated for the electronic propagation. The starting and final geometries of the run trajectories are provided in Supplementary Data 1.

## Data availability
Source data are provided with this paper.

## Code availability
The code that supports the findings of this study is available from the corresponding authors upon request.

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

## Acknowledgements
We acknowledge financial support from the European Research Council under the ERC-2014-StG STARLIGHT (grant no. 637756), the Cluster of Excellence 'CUI: Advanced Imaging of Matter' of the Deutsche Forschungsgemeinschaft (DFG) - EXC 2056 – project ID 390715994, the German Research Foundation (DFG)—SFB-925—project 170620586 and the Open Access Publication Fund of Universität Hamburg. V.W. acknowledges support from the Hamburg Partnership for Innovation, Education and Research (PIER) (PIF-2021-03). A.T. acknowledges support from the Helmholtz Association under the Helmholtz Young Investigator Group VH-NG-1603, and financial support from the European Research Council under the ERC SoftMeter no. 101076500. Views and opinions expressed are however those of the author(s) only and do not necessarily reflect those of the European Union or the European Research Council Executive Agency. Neither the European Union nor the granting authority can be held responsible for them. All calculations were performed at the Mare Nostrum Supercomputer of the Red Española de Supercomputación (BSC-RES) and the Centro de Computación Científica de la Universidad Autónoma de Madrid (CCC-UAM). This paper is based upon work from COST Action CA18222 (AttoChem), supported by COST (European Cooperation in Science and Technology) and has been supported by the projects PID2019-105458RB-I00, PID2019-106732GB-I00, PID2022-138288NB-C31, and PID2022-138288NB-C32 funded by MCIN/ AEI /10.13039/501100011033 and by the European Union "NextGenerationEU" / PRTRMICINN programs, and the "Severo Ochoa" Program for Centers of Excellence in R&D (CEX2020-001039-S). Finally, we acknowledge fruitful discussions with Prof. Luis Bañares.

## Author contributions
F.M. and F.C. supervised the project. L.C., S.R., K.S., E.M., V.W., and A.T. performed the measurements. L.C. and E.M. performed the data analysis. J.G.V. performed the simulation. L.C., E.M., J.G.V., F.M., and F.C. wrote the manuscript. All authors contributed to the discussion of the results and the editing of the manuscript.

## Funding

## Competing interests
The authors declare no competing interests.
