## [Peer Review File · Nature Communications]

Few-femtosecond time-resolved study of the UV-induced dissociative dynamics of iodomethaneREVIEWER COMMENTS

Reviewer #1 (Remarks to the Author):

This manuscript presents the results of a time resolved photoion spectroscopy experiments employing few fs UV and near IR laser pulses. The authors excite a dissociative wave packet in the molecule CH₃I and probe the time evolution of the wave packet by ionizing the molecule and detecting ionic fragments. The measurements are compared with semiclassical (SHARC) dynamics calculations which effectively capture the non-adiabatic excited state dynamics but do not describe the ionization of the molecule.

The dissociation of halogen substituted methanes is a well studied topic, and time resolved photoionization spectroscopy is a well established approach that has been employed by many research groups, including that of Werner Fuss, who used multiphoton ionization to track the non-adiabatic excited state dynamics of many molecules (including cyclohexadiene with 13 fs time resolution - Kosma, K., et al. "Cyclohexadiene ring opening observed with 13 fs resolution: coherent oscillations confirm the reaction path." *Physical Chemistry Chemical Physics* 11.1 (2009): 172-181.). One of the main difficulties with this approach is that multiphoton ionization is very difficult to treat theoretically, and therefore it is not generally possible to model the measurement observable quantitatively. This allows for qualitative interpretations of the dynamics, but makes it difficult to get detailed insight or direct evidence of specific dynamics because it is difficult to disentangle the excited state dynamics from the variation in the multiphoton ionization rate with nuclear coordinate. Multiphoton resonant enhancement of the ionization yield can for instance have a significant effect on the ionization yield, and without detailed information on when and how much this effects the ionization, it is not clear to me how one can unambiguously interpret the ionization yield vs pump probe delay. There have been a few papers published on the difference between multiphoton and single photon ionization, and the general conclusion is that they yield qualitatively similar results, but not quantitatively matching (see for example Koch, Markus, Thomas JA Wolf, and Markus Gühr. "Understanding the modulation mechanism in resonance-enhanced multiphoton probing of molecular dynamics." *Physical Review A* 91.3 (2015): 031403.). This is at least part of the reason for the difficulty in comparing experiment and theory in figure 3. I found it quite difficult to interpret this figure, and I did not find any compelling agreement between experiment and theory. I think that the authors struggle in their discussion of this figure to provide a clear interpretation of the measurements, and I think that this is because there is no quantitative calculation of the actual measurement observable (field free dynamics plus the ionization process). Because the comparison between experiment and theory is not very satisfying, I feel that the observation of the wave packet motion through the CI is not very direct. The dynamics calculations are compelling, but I don't think that the comparison with the measurements justifies the statement: "In agreement with the experiment, the model accurately identifies the

main dissociation channels and further reveals a clear signature of the conical intersection in the time-dependent yield of the iodine ion." I was not convinced that the measurements reveal a clear signature of the CI, and note that there have been many other measurements that yield more direct evidence of non-adiabatic dynamics.

In conclusion, while I think that the time resolution achieved by the authors is impressive, and the calculations are solid, I do not think that their combination provides a significant enough development to merit publication in *Nature Communications*.

Reviewer #2 (Remarks to the Author):

In the manuscript Few-femtosecond time-resolved study of the UV-induced dissociative dynamics of iodomethane, the authors present ultrafast UV-pump NIR-probe data as well as results from non-adiabatic molecular dynamics simulations using the program SHARC to elucidate the mentioned dynamics. We appreciate the unprecedented short duration of the pulses, which makes most of the dynamics field free, as well as the combination of a state-of-the-art experiment with modern computer simulation methods. Many interesting points are made. However, we do not think that the analysis presented takes full advantage of this combination, and the presentation

should be improved in several respects. We have the following questions that must be addressed before the manuscript can be considered for publication.

1) The simulations:

A) It is very hard to fully comprehend the simulation of the ionization processes from the main text. We suggest including the first six lines of the second paragraph on page 22 (page 7 of the SI) in the main text (or at least refer to them from the text). In the first phrase of this paragraph, the wording "the trajectory count in the energy region between $n-1$ and n NIR photon energies" is unclear. We assume that it means counting the trajectories with the electronic (potential, if you will) energy difference between the relevant ionic and neutral states (i.e., the relevant ionization threshold) between $n-1$ and n NIR photon energies. Please clarify the phrase.

B) According to the caption to Fig. 3, the figure "shows the number of trajectories contributing to the respective ions creation without considering ionization probabilities". Similar wording is used elsewhere. However, cf., the comment in item A, the simulation procedure seems to neglect multi-photon above threshold ionization <https://doi.org/10.1038/s41598-021-89733-z>. Otherwise, all fragments would be produced by the number of photons shown in Fig. 3 (and S5-6) and all higher numbers. It would be helpful to the reader if this was stated clearly.

C) How does the simulation of the ionization processes discriminate between the different excitation channels, e.g. in the asymptotic region, where all electronic (potential) energy curves are 'flat'? From Fig. 3 (a-c) it seems that all the trajectories have reached the asymptotic region after 40 fs (on page 9 it says "after 30 fs"). This implies that all trajectories moving on the same neutral state have the same potential (electronic) energy and, therefore, the same ionization thresholds. Figure S6 shows that 6-photon probe excitation in this region can produce both I^+ or CH_3^+ from either of the two neutral states. Then, how come most of the trajectories when probing with 6 NIR photons after 40 fs lead to I^+ according to Fig. 3 (a-c)? Similarly, how come there are different (non-zero) probabilities leading to I^+ when probing with 6 or 7 (or 8) NIR photons after 40 fs? Please elucidate in the manuscript.

D) References to program SHARC should follow the guideline https://sharc-md.org/?page_id=286, and references to other studies of dissociative dynamics using SHARC could be included in the manuscript, e.g., <https://doi.org/10.1039/d1cp00771h>.

2) Figures:

A) In Fig. 1 please explain in the caption the dashed blue, red, and orange curves.

B) In Fig. 2 it would be helpful to the reader if all three panels used the same unit on the y-axes (as in Fig. S3) and if an explaining of the unit was given.

C) In Fig. 5 all four molecules are depicted with the same C-I bond length which is quite confusing. Please change the figure to reflect that the dissociating (upper right) molecule has an extended (or no) C-I bond, while the other three have the equilibrium C-I bond length.

D) In Fig. S3 the y-axes should be switched between the first and last panel to conform with Fig. 2.

E) Suggestion for a new figure: It would be very helpful to the reader including a figure illustrating the C-I bond-length distribution in the swarm of trajectories as a function of time, e.g., plotting all the trajectories or showing a 2D color plot like in Fig. 4 with time on the x-axis and C-I bond length on the y-axis.

3) Data analysis:

A) The curve-fitting model composed of three terms used in Fig. 2 seems rather arbitrary. Is there a good physical reasoning for this model?

B) The manuscript presents both experimental data and computer simulation results. The authors conclude by comparison that the characteristic minimum observed in the measured I^+ yield at around 20-30 fs implies that the majority of the trajectories go through the conical intersection (adiabatic dynamics) and that I^+ is produced (predominantly) by absorption of 7 probe photons. Can the former also be concluded from the simulations alone by calculation of the probe-free (branching) ratio between the number trajectories ending in the ground state of iodine I and the number trajectories ending in the excited state I^* ?

C) Returning to item A: Instead of the curve-fitting model by three terms used in Fig. 2, could the solid lines in Fig. 3 for the different excitation channels be used? For instance, for the I^+ production, the 5-photon curve has a similar shape as the decaying fitting term, while the 6- and 7-photon curves look somewhat like the long-lived contribution if the latter was given a non-zero

value at early times. Likewise, the 5- and 6-photon curves for the CH₃⁺ production (apart from the delayed onset) look like decaying and long-lived fitting terms, respectively, especially keeping in mind that the ratio between the 5- and 6-photon curves should be increased when considering that 5-photon absorption is more likely than 6-photon absorption. It would increase the impact of this work if (more) speculations along these lines were included in the manuscript.

D) Further speculations: It is concluded from comparing the shapes of the experimental signals and the calculated curves that CH₃I⁺ is produced from 3-photon absorption, CH₃⁺ from 5- and 6-photon absorption, and I⁺ from 5- and 7-photon absorption. Considering that low-photon absorption is more likely than high-photon absorption, these conclusions would imply (everything else being equal) that the CH₃I⁺ signal should be stronger than the CH₃⁺ signal which, in turn, should be stronger than the I⁺ signal. This is, in fact, the case for the signals in Fig. 2. Maybe that is worth a comment in the manuscript.

4) Editorial comments:

The manuscript and especially the SI could benefit from proof reading. There are unexplained abbreviations such as "SH-FROG" in the Experimental setup on page 16 (page 1 of the SI) and the Experimental Setup needs more references. Mismatch between Fig. 2 and Fig. S3 is already mentioned above. The figure references are wrong on page 19 (page 4 of the SI) where S2 should be S3 and S3 should be S4.

5) Impact and interest for the general readership:

Iodomethane is a popular prototype molecule, and the work will impact the field of femtochemistry of small gas-phase molecules because of 1) the identification of signatures of the conical intersection between the neutral states because of the due to the field-free propagation at the end of the UV pulse, and 2) the showcasing of an ultrafast stabilization scheme against dissociation by few-femtosecond ionization. For the general readership (within femtochemistry) the short UV pulse will be appreciated, and so would be a scheme for benchmarking of trajectory calculations (to cite the abstract). However, as pointed out above, to achieve this, the trajectory calculations must be described in more detail.

Reviewer #3 (Remarks to the Author):

Manuscript ID: NCOMMS-23-62707

Title: Few-femtosecond time-resolved study of the UV-induced dissociative dynamics of iodomethane

Dear Editor,

Thank you for inviting us to revise our work for further consideration in your journal. We have now addressed all the points raised by the reviewers and we have done a significant effort not only to provide additional details on the theoretical model but also to better describe the multiphoton probing step.

We sincerely hope that the manuscript is now suitable for publication in Nature Communications and we look forward to your positive feedback.

Sincerely,

List of changes:

In reply to the Reviewers' comments, we have revised the manuscript and introduced the following changes.

Main text:

- Abstract: we mitigated the sentence regarding the signature of the conical intersection.
- Figure 1 and its caption have been updated.
- Figure 2 has been corrected with the same unit for all the y-axes and the caption has been improved with the description of the ion yield unit.
- Page 5: a sentence has been introduced to describe the relative yield of the observed ions.
- Page 7-8:
 - We have been working on the text to improve the description of the employed theoretical model.
 - Caption of Figure 3 has also been improved.
 - We included a short description of the procedure followed to extract the probability of producing a specific ion from the calculated trajectories for each multiphoton channel.
- Page 8: we introduced a sentence about the role of resonances when using multiphoton ionization as a probe of the molecular dynamics.
- Page 11:
 - We include the role of the probe-free branching ratios in favoring seven-photon absorption.
 - We also included an additional paragraph to describe a best-fit procedure used to improve our comparison between the extracted n-photon theory curves and the experimental data.
 - We compare the assigned multiphoton channels with the relative amplitude of the different ion yield.
- Page 12: we included an additional paragraph describing a new model, which includes an approximate solution of the time dependent Schrödinger equation, that we have been introducing to confirm our conclusions.
- In Figure 5 no C-I bond is represented in the neutrally dissociated molecule (upper right panel).
- Page 16: we mitigate our statement regarding the signature of the conical intersection.

Supplementary Information:

- We explicitly explain the acronym SH-FROG and added a reference.
- Figure S3 reports the revised panels for the three ions.
- Page 19: we included a description of the units used for the ion yield.
- Page 20: corrected typos regarding Fig. S2, S3, S4 referencing.

- Page 22: we introduced a new sentence and with references to better describe the curve-fitting model.
- Page 23:
 - We clarify the method used for calculating the dynamics.
 - We added a subsection for visualizing the dissociative trajectories in the neutral molecule, with a new figure S5.
- Page 26: we introduced a subsection describing the attempt of assigning a relative weight to the n-photon channels by fitting the experimental data with a linear combination of the theoretically extracted curves. The resulting fitting coefficients are reported in a new table.
- Page 27: we introduced a subsection describing an additional model, which tries to include the multiphoton probing step through an approximate solution of the Time-Dependent Schrodinger Equation. The results of the calculation are shown in a new figure.

We are grateful to all Reviewers for their work and useful comments. We are confident that the new version of the manuscript fully addresses their concerns and meets the criteria for publication in Nature Communications.

In the following, we report a detailed response to the Reviewers' comments.

Blue: reviewers' comment
Red: change in manuscript
Black: authors' answer

REVIEWER COMMENTS

Reviewer #1 (Remarks to the Author):

This manuscript presents the results of a time-resolved photoion spectroscopy experiments employing few fs UV and near IR laser pulses. The authors excite a dissociative wave packet in the molecule CH₃I and probe the time evolution of the wave packet by ionizing the molecule and detecting ionic fragments. The measurements are compared with semiclassical (SHARC) dynamics calculations which effectively capture the non-adiabatic excited state dynamics but do not describe the ionization of the molecule.

The dissociation of halogen substituted methanes is a well studied topic, and time resolved photoionization spectroscopy is a well established approach that has been employed by many research groups, including that of Werner Fuss, who used multiphoton ionization to track the non-adiabatic excited state dynamics of many molecules (including cyclohexadiene with 13 fs time resolution - Kosma, K., et al. "Cyclohexadiene ring opening observed with 13 fs resolution: coherent oscillations confirm the reaction path." *Physical Chemistry Chemical Physics* 11.1 (2009): 172-181.). One of the main difficulties with this approach is that multiphoton ionization is very difficult to treat theoretically, and therefore it is not generally possible to model the measurement observable quantitatively. This allows for qualitative interpretations of the dynamics, but makes it difficult to get detailed insight or direct evidence of specific dynamics because it is difficult to disentangle the excited state dynamics from the variation in the multiphoton ionization rate with nuclear coordinate. Multiphoton resonant enhancement of the ionization yield can for instance have a significant effect on the ionization yield, and without detailed information on when and how much this effects the ionization, it is not clear to me how one can unambiguously interpret the ionization yield vs pump probe delay. There have been a few papers published on the difference between multiphoton and

single photon ionization, and the general conclusion is that they yield qualitatively similar results, but not quantitatively matching (see for example Koch, Markus, Thomas JA Wolf, and Markus Gühr. "Understanding the modulation mechanism in resonance-enhanced multiphoton probing of molecular dynamics." *Physical Review A* 91.3 (2015): 031403.).

This is at least part of the reason for the difficulty in comparing experiment and theory in figure 3. I found it quite difficult to interpret this figure, and I did not find any compelling agreement between experiment and theory. I think that the authors struggle in their discussion of this figure to provide a clear interpretation of the measurements, and I think that this is because there is no quantitative calculation of the actual measurement observable (field free dynamics plus the ionization process). Because the comparison between experiment and theory is not very satisfying, I feel that the observation of the wave packet motion through the CI is not very direct. The dynamics calculations are compelling, but I don't think that the comparison with the measurements justifies the statement: "In agreement with the experiment, the model accurately identifies the main dissociation channels and further reveals a clear signature of the conical intersection in the time-dependent yield of the iodine ion." I was not convinced that the measurements reveal a clear signature of the CI, and note that there have been many other measurements that yield more direct evidence of non-adiabatic dynamics.

In conclusion, while I think that the time resolution achieved by the authors is impressive, and the calculations are solid, I do not think that their combination provides a significant enough development to merit publication in *Nature Communications*.

We thank the reviewer for providing insightful feedback. We appreciate the recognition of the challenges associated with time-resolved photoion spectroscopy, particularly in capturing the quantitative aspects of the observed dynamics when using a multiphoton probe. Indeed, the complexity of multiphoton ionization makes both the interpretation of the experimental data and the direct comparison with theoretical calculations more difficult than when using single photon probes.

Once again, we would like to stress that the unique combination of ultrabroadband UV pulses with short NIR pulses provides unprecedented time resolution to follow the very fast (a few tens of femtoseconds) dissociative dynamics of iodomethane in a pump-free scenario, as described in the main text, and we still see this as a major breakthrough. Additionally, our detailed analysis of the dissociative trajectories offers an interesting approach to disentangle the various multiphoton contributions in this benchmark molecule, and we could foresee the extension of this method to other systems in the future.

We certainly agree with the referee that multiphoton resonant enhancement of the ionization yield cannot be excluded a priori. However, iodomethane has been intensively investigated and in reference [\[https://doi.org/10.1039/D3SC04065H\]](https://doi.org/10.1039/D3SC04065H) it is shown that even at binding energy 12-14 eV no significant increase of the fragment yields is produced. Additionally, in reference [\[https://doi.org/10.1063/5.0161628\]](https://doi.org/10.1063/5.0161628) the role of resonances is investigated in the FC region and, in the case of an 800-nm pulse, it is demonstrated that REMPI via the 6p Rydberg is only enabled when a stark shift of the electronic states is occurring due to the high-intensity pulse. In our case, if we consider both the UV pump and the IR probe pulse there is no significant energy overlap with the identified Rydberg states (see Fig. R1) and, considering that our intensity is considerably lower than in reference [\[https://doi.org/10.1063/5.0161628\]](https://doi.org/10.1063/5.0161628), we don't expect any significant REMPI process to occur in the FC region.

Figure R1: (Solid line) Rydberg states spectrum obtained in synchrotron from [<https://doi.org/10.1063/5.0161628>]. In red area the infrared probe spectrum used in the current work, shifted in energy by 4.95 eV, corresponding 1 UV photon excitation.

Although we cannot entirely exclude the presence of resonances outside the FC region, in a REMPI scheme we would expect a substantial increase in ion yield at longer delays compared to early time delays, which is not observed in the experimental data.

To clarify this point, we included the following paragraph in the manuscript:

“We note that our theoretical approach cannot model resonance-enhanced multi-photon ionization (REMPI). Although detailed knowledge about the resonances is in general needed for a proper interpretation of transient signals of molecular dynamics [doi.org/10.1103/PhysRevA.91.031403], previous works on iodomethane [doi.org/10.1063/5.0161628, doi.org/10.1063/1.3236808] indicate that REMPI does not play a relevant role in our experimental conditions.”

Full calculations of the multiphoton ionization step for a molecule like CH₃I, even for a single pump-probe delay is a formidable challenge, which is currently inaccessible. However, in the attempt to get more quantitative information as suggested by the reviewer, we have performed new calculations in which the multiphoton probe step is approximately described. The calculations are based on the solution of the Time Dependent Schrödinger Equation (TDSE), in which the NIR-induced dipole couplings between the bound electronic states of the neutral molecule shown in Fig. 1 and between the electronic states of the molecular cation, respectively, are explicitly included, while the couplings between those bound states and the electronic continuum states associated with the various dissociative channels of the molecular cation are approximately represented by the corresponding Dyson norms after introducing a common rescaling factor to account for the multiphoton nature of the corresponding ionizing transitions. We note that by restricting the number of bound states of the neutral molecule to those shown in Fig. 1 of the main text, we are ignoring the possible effect of the Rydberg states in the multiphoton ionization process. However, since dipole transition moments associated with these states rapidly decrease as they approach the ionization limit, we do not expect they will play a significant role. The results we have obtained by using this more elaborate approach are in qualitative agreement with those obtained from the simpler model reported previously in the manuscript, thus reinforcing our initial interpretation. The details and results of these new calculations are now summarized in the main text and presented in more detail in the SI.

We understand the referee’s concerns concerning the interpretation of Figure 3 in terms of the much simpler model described in the original version of the manuscript. However, we believe that the simplicity of such model allows for a clearer understanding of the different aspects of this complicated process. We would like to emphasize that we find extremely remarkable that, even under these extreme simplifications, the simulated curves still reproduce qualitatively the dynamics observed in the experiment. For this reason, in the main text we have maintained the original discussion in terms

of this simple model with the necessary improvements and we refer now to the SI for the more elaborate TDSE calculations described above to support the analysis.

Finally, we agree with the referee that there are other measurements performed in CH_3I that reveal more direct signatures of the CI, however none has been performed with the time resolution required to follow the passage through the CI in a pump-free condition. We have now mitigated our statement with:

"In agreement with the experiment, the model allows for the identification of the main dissociation channels and indirectly reveals a signature of the conical intersection in the time-dependent yield of the iodine ion."

Reviewer #2 (Remarks to the Author):

In the manuscript Few-femtosecond time-resolved study of the UV-induced dissociative dynamics of iodomethane, the authors present ultrafast UV-pump NIR-probe data as well as results from non-adiabatic molecular dynamics simulations using the program SHARC to elucidate the mentioned dynamics. We appreciate the unprecedented short duration of the pulses, which makes most of the dynamics field free, as well as the combination of a state-of-the-art experiment with modern computer simulation methods. Many interesting points are made. However, we do not think that the analysis presented takes full advantage of this combination, and the presentation should be improved in several respects. We have the following questions that must be addressed before the manuscript can be considered for publication.

1) The simulations:

A) It is very hard to fully comprehend the simulation of the ionization processes from the main text. We suggest including the first six lines of the second paragraph on page 22 (page 7 of the SI) in the main text (or at least refer to them from the text). In the first phrase of this paragraph, the wording "the trajectory count in the energy region between $n-1$ and n NIR photon energies" is unclear. We assume that it means counting the trajectories with the electronic (potential, if you will) energy difference between the relevant ionic and neutral states (i.e., the relevant ionization threshold) between $n-1$ and n NIR photon energies. Please clarify the phrase.

We apologize for the lack of clarity in the description of the ionization step in our simulations. We have made an effort to improve this description in the main text (see the revised version). We have also added a new paragraph following the suggestion of the reviewer. Here we only reproduce the latter:

"Finally, at each time step, we associate a trajectory leading to an ion i with a specific number of absorbed NIR photons n , when such trajectory lies in the energy range between $n-1$ and n NIR photons. Then, we assume that the probability of producing an ion i at a given time after the absorption of n photons is simply given by the total count of trajectories leading to the ion i in the corresponding energy region. In doing so, we assume that the ionization cross sections are independent of the photoelectron energy, and they are identical for all final states of the cation. For trajectories with multiple dissociation channels open, we assign it only to the ion fragment with the highest threshold."

We have also improved the description reported on page 25 of the SI.

B) According to the caption to Fig. 3, the figure "shows the number of trajectories contributing to the respective ions creation without considering ionization probabilities". Similar wording is used

elsewhere. However, cf., the comment in item A, the simulation procedure seems to neglect multiphoton above threshold ionization <https://doi.org/10.1038/s41598-021-89733-z>. Otherwise, all fragments would be produced by the number of photons shown in Fig. 3 (and S5-6) and all higher numbers. It would be helpful to the reader if this was stated clearly.

We agree with the reviewer that this sounds a bit contradictory. We have modified the text as described in our answer to point A.

C) How does the simulation of the ionization processes discriminate between the different excitation channels, e.g. in the asymptotic region, where all electronic (potential) energy curves are 'flat'? From Fig. 3 (a-c) it seems that all the trajectories have reached the asymptotic region after 40 fs (on page 9 it says "after 30 fs"). This implies that all trajectories moving on the same neutral state have the same potential (electronic) energy and, therefore, the same ionization thresholds. Figure S6 shows that 6-photon probe excitation in this region can produce both I⁺ or CH₃⁺ from either of the two neutral states. Then, how come most of the trajectories when probing with 6 NIR photons after 40 fs lead to I⁺ according to Fig. 3 (a-c)? Similarly, how come there are different (non-zero) probabilities leading to I⁺ when probing with 6 or 7 (or 8) NIR photons after 40 fs? Please elucidate in the manuscript.

We understand that, without further explanations, the observed variations in the I⁺ probabilities after 40 fs may be unexpected. In the dissociation limit, both CH₃ and I/I* fragments will be ionized and the resulting CH₃⁺ and I⁺ ionic fragments need to be correctly assigned to the different fragmentation channels. The case of I and I* is very simple: since the photoelectron can take any angular momentum, all possible transitions from here to any channel dissociating into I⁺ need to be considered. In the case of CH₃, ionization leads to CH₃⁺ while maintaining I and I* unchanged, so, a priori, there should be no ambiguity either. However, the ordering of the channels involving I and I* can change during the dynamics and this could be a problem for the identification of the correct dissociation limit. Encouraged by the referee's comment, and to check that a potential mistake in ordering the various channels is not the reason for the variations observed beyond 40 fs, we have performed new calculations in which the multiphoton probe step is approximately described. The calculations are based on the solution of the Time Dependent Schrödinger Equation (TDSE), in which the NIR-induced dipole couplings between the bound electronic states of the neutral molecule shown in Fig. 1 of the main text and between the electronic states of the molecular cation, respectively, are explicitly included. The couplings between those bound states and the electronic continuum states, associated with the various dissociative channels of the molecular cation, are approximately represented by the corresponding Dyson norms after introducing a common rescaling factor to account for the multiphoton nature of the corresponding ionizing transitions. We note that by restricting the number of bound states of the neutral molecule to those shown in Fig. 1 of the main text, we are ignoring the possible effect of the Rydberg states in the multiphoton ionization process, however since dipole transition moments associated with these states rapidly decrease as they approach the ionization limit, we do not expect they will play a significant role. We also note that in this formalism, the Dyson norm connecting the I + CH₃ and I* + CH₃⁺ dissociation limits is zero (the electronic states differ by two electrons). The same is true for the connection between the I* + CH₃ and I + CH₃⁺ dissociation limits, so that there is no ambiguity in the assignment of the channels involving ionization from I and I*. The results we have obtained by using this more elaborate approach are in qualitative agreement with those obtained from the simpler model reported previously in the manuscript, thus reinforcing our initial interpretation. The details and results of these new calculations are now summarized in the main text and presented in more detail in the SI.

D) References to program SHARC should follow the guideline https://sharc-md.org/?page_id=286, and references to other studies of dissociative dynamics using SHARC could be included in the manuscript, e.g., <https://doi.org/10.1039/d1cp00771h>.

We would like to clarify that we are not using the sharc-md code (the original SHARC paper [ref 20] does not refer to that code either). Instead, we are using a modified version that we have made available for the referee in the repository quoted in ref. 21. We apologize if the text was not clear. To avoid any confusion, we have modified it as follows: “the previous protocol was connected with the SHARC^{20,21} method”, avoiding mentioning any program. Concerning the use of the SHARC methodology for dissociation dynamics, in addition to references 11, 16 and 18 already cited in the original version of the manuscript, we have now included the reference suggested by the referee.

2) Figures:

A) In Fig. 1 please explain in the caption the dashed blue, red, and orange curves.

We thank the reviewer for noticing the missing descriptions for these curves. The dashed lines for the potential energy curves in Fig. 1 represent curves that were calculated but are not relevant in our experiment. For example, the blue and red dashed lines in the neutral represent states for which the transition dipole is negligible [ref. Alekseyev 2007]. The UV single photon absorption mainly populates the 3Q_0 state and the 1Q_1 state through the conical intersection. For the cationic states, we used solid lines for the states, which we believe are the main responsible for the experimentally observed ions. Following this idea, considering that we partially assign the iodine signal to a 7-photon transition, even states of higher energy than A^2A_1 needs to be highlighted with solid lines.

We have replaced in Fig. 1 caption the following.

“The most relevant states are highlighted with solid curves together with their dissociation products.”

with

“The states responsible for the observed excited state dynamics are highlighted with solid lines, along with their dissociation products. Dashed lines are used to indicate states included in the calculation, which negligibly contribute to the observed dynamics.”

B) In Fig. 2 it would be helpful to the reader if all three panels used the same unit on the y-axes (as in Fig. S3) and if an explaining of the unit was given.

We thank the referee for the comment. We updated Fig. 2 to have the same units on the y-axis for all the panels and expanded the caption with “The ion yield is reported as amplified charge in pC (recorded by the acquisition system) per laser shot (see SI for details).” As the analysis and conclusions do not depend on comparing any absolute signal strengths, we defer the precise description of the acquisition system and shown quantity to the supplementary information. On page 19, the previous description has been expanded with the following: “The recorded voltage is divided by the 50 Ω characteristic impedance to obtain a current. The current’s integral over the short time interval corresponding to a specific ion in the time-of-flight mass spectrum gives the detected charge. In Figure 2 we show the average charge per laser shot in picocoulombs with the units expressed as ‘pC/shot’.”

C) In Fig. 5 all four molecules are depicted with the same C-I bond length which is quite confusing. Please change the figure to reflect that the dissociating (upper right) molecule has an extended (or no) C-I bond, while the other three have the equilibrium C-I bond length.

We thank the reviewer for the suggestion. We have now split the two fragments along the C-I bond in the upper right molecule, indicating dissociation upon UV excitation.

D) In Fig. S3 the y-axes should be switched between the first and last panel to conform with Fig. 2.

We deeply thank the reviewer for this close examination that revealed the mistake. It was not the axes but the entire panels which have been labelled with the wrong ion.

E) Suggestion for a new figure: It would be very helpful to the reader including a figure illustrating the C-I bond-length distribution in the swarm of trajectories as a function of time, e.g., plotting all the trajectories or showing a 2D color plot like in Fig. 4 with time on the x-axis and C-I bond length on the y-axis.

Following the referee's recommendation, we have included a new Figure in the SI with the following text:

“Figure R2 shows the evolution of the average C-I distance for two sets of trajectories. In the first set, the average has been performed by considering the electronic states leading to dissociation into I (the 8 lowest states) and, in the second set, by considering the states leading to dissociation into I* (the 4 upper states). One can see two different evolutions: the faster one corresponds to dissociation into the I channels, whereas the slower one to dissociation into the I* channels. In the case of I*, a clear change of the slope at around 40 fs is observed (when a change in the I+ channel is also observed experimentally). At short times, the bifurcation of the two channels, related to the passage through the conical intersection, is quite apparent at around 14.5 fs. This value is very similar to those obtained in refs. 13 (15 fs) and 16 (13 fs), but it is larger than that directly obtained from the energy difference between the states (10.5 fs). This 4-fs difference (between 10.5 fs and 14.5 fs) can be explained as the time required for the ensemble of trajectories to experience the CI-induced gradient and separate.”

Figure R2: Weighted convolution of the C-I distance for all the trajectories as a function of time. The convolution was done with a Gaussian function of 0.1 Angstroms FWHM. On top of the convoluted data, the weighted average distance when including the trajectories in the first 8 (red) and the last 4 (cyan) electronic states is

also included. The inset focus on the first 30 fs, where the bifurcation between the two channels I (red) and I* (cyan) can be observed.

3) Data analysis:

A) The curve-fitting model composed of three terms used in Fig. 2 seems rather arbitrary. Is there a good physical reasoning for this model?

Fitting a time-resolved signal often implies the use of more than one term, when more than one transition or population contribute to the observable. To provide more references about this conventional curve fitting model, we inserted the following sentence in the corresponding supplementary Information section:

“Similar curve-fitting models with time offsets, representing the time required by the wavepacket to reach a precise transition point, have for instance been used in Ref. ^{14, 19, 35}”

B) The manuscript presents both experimental data and computer simulation results. The authors conclude by comparison that the characteristic minimum observed in the measured I+ yield at around 20-30 fs implies that the majority of the trajectories go through the conical intersection (diabatic dynamics) and that I+ is produced (predominantly) by absorption of 7 probe photons. Can the former also be concluded from the simulations alone by calculation of the probe-free (branching) ratio between the number trajectories ending in the ground state of iodine I and the number trajectories ending in the excited state I*?

Following the reviewer’s suggestion we have calculated the probe-free I*/(I+I*) ratio by integrating the result of Fig R2. We have obtained $I^*/(I+I^*) = 0.62$. This result is compatible with those obtained in previous works, which lie between 0.47 and 0.82 depending on the quantum state of the CH₃ product [DOI:10/gmqdkr]. So, although the production of I* is dominant, that of I is also important, so that ionization of both I* and I by the NIR pulse significantly contribute to the formation of I+. Therefore, to understand why I+ is mainly formed by absorption of 7 probe photons, one should consider both the I and I* channels. The key factor is the number of I+ final states accessible after absorption of n NIR photons. This number is given in the following Table:

No. of NIR photons	Initial state	No. of accessible I+ states	Total no. of accessible I+ states
6	I	0	5
	I*	5	
7	I	8	17
	I*	9	
8	I	1	6
	I*	5	

As can be seen, the number of accessible I+ states is more than three times larger for absorption of 7 photons than for absorption of 6 photons. Hence, one can reasonably expect that the 7-photon absorption process will have an important contribution to the I+ yield even though state-to-state transitions induced by absorption of 6 photons should in principle be more likely. This is the reason why the 7-photon curve dominates at long times in Figure 3e.

To clarify this point we have added the sentences on page 11 of the main manuscript:

“Absorption of 7 NIR photons at the longer time delays results as the dominant process because (i) the probe-free $I^*/(I+I^*)$ and $I/(I+I^*)$ branching ratios are 0.68 and 0.32, respectively, so that I^+ can be produced by ionizing both I and I^* , and (ii) the total number of accessible I^+ states from I and I^* is more than three times larger for seven-photon absorption than for six-photon absorption.”

C) Returning to item A: Instead of the curve-fitting model by three terms used in Fig. 2, could the solid lines in Fig. 3 for the different excitation channels be used? For instance, for the I^+ production, the 5-photon curve has a similar shape as the decaying fitting term, while the 6- and 7-photon curves look somewhat like the long-lived contribution if the latter was given a non-zero value at early times. Likewise, the 5- and 6-photon curves for the CH_3^+ production (apart from the delayed onset) look like decaying and long-lived fitting terms, respectively, especially keeping in mind that the ratio between the 5- and 6-photon curves should be increased when considering that 5-photon absorption is more likely than 6-photon absorption. It would increase the impact of this work if (more) speculations along these lines were included in the manuscript.

We appreciate the reviewer's suggestion. The fit reported in Fig. 2 and described in the results section represent an all-experimental approach to extract timing information from the data. As discussed at point 3A, we attempted to integrate curves typically used in pump-probe spectroscopy to decompose our complex ion signals. Additionally, considering the qualitative agreement we observe when overlapping the theoretical curves with the experimental data, we used the theoretical results to try to decompose the various multiphoton contributions. However, we would like to stress that this comparison needs to stay at the quantitative level because the theoretical curves only account for number of trajectories, without including the cross section of the multiphoton transitions. A different approach which tries to include the multiphoton probing by evaluating the Dyson norms is described in point 1C. Nevertheless, triggered by the reviewer's suggestion, we tried to explore a different approach where the theoretically retrieved curves are weighted to best fit the experimental curves. The extracted fitting coefficients should then reflect the relative contribution of the multiphoton transitions to the observed ion signal.

Figure R3 shows the result. For each ion, the best fit (solid red line) is obtained by a linear combination of the normalized set of nine curves reported in Fig. S5.

Figure R3: Best fitting curves (red solid lines) obtained as linear combination of the theoretical curves from Fig. S5. Dots indicate experimental data points (as reported in Fig. 2 of the main manuscript), while shaded area indicate the standard deviation. The individual n -photons theoretical curves weighted by their relative coefficients extracted from the fit are also reported here as coloured solid lines.

Parameters	CH_3I^+	I^+	CH_3^+
A_0	4.6 (fixed)	0.014 (fixed)	0.081 (fixed)
A_3	9.3	0.1	0
A_4	5.5	0	$2.12e-07$
A_5	10	0.038	$4.89e-06$
A_6	0	0.002	$2.91e-06$
A_7	0	0.015	0
A_8	0	$8.6e-06$	0

Table R1: Coefficients obtained by fitting the set of normalized curves from figure S5 to the experimental data.

Table R1 reports the result for the coefficients. Overall, the fit confirms the assignment suggested in the main text. For the parent, the observed fast decay can only be properly fitted with the 3-photon curve, which reproduces the overall trend. When considering the I^+ fragment, many more curves can contribute to the observed signal, however the fitting clearly suggests which curves follow better the experimental results. In particular, we observe that only the coefficients associated to the 5 and 7 photon transitions are not negligible. This confirms our previous conclusion that the six-photon transition only marginally contribute to the iodine signal. Finally, for the methyl fragment, only the

coefficient associated to the 6 and the 7 photon transitions contribute to the long-lived component of the data, aligning with our previous interpretation. To conclude, when using a linear combination of the n -photon theoretical curves to fit the experimental data we numerically confirm our previous assignments based on the temporal shape of the curves. However, we consider this as a more refined version of what we already stated in the main text. The main issue here arises by the fact that the fitting is done with two different physical quantities which in principle cannot be compared as they are, and their relationship in between would require calculations of the full multi-photon probing steps. In this context, as mentioned above, we made an additional effort of addressing the relative weight of the various multiphoton contributions by using a different model based on the solution of the time-dependent Schrödinger equation. Since the results of this model are very similar to those reported in the original version of the manuscript, we decided to include the details of the above analysis in the supplementary material, sub-section "Relative contribution of the theoretically extracted probabilities".

In the main text, at the end of the first paragraph of the Discussions we added:

"While direct comparison between theoretical curves and experimental data is not straightforward, these curves still offer valuable insights. We used a best-fit approach to estimate the contribution of each theoretical curve to the experimental signal for every ion species by assuming that the experimental signal is represented by a linear combination of these curves. To this goal, all the curves have been convoluted with the IRF to include the resolution given by the finite duration of our pulses. Detailed results can be found in the Supplementary Information (SI). For the parent ion, the only coefficient that is significantly different from zero is that of the three-photon curve. For the methyl ion, a combination of only five- and six-photon curves best fits the experimental data. The most interesting case is the one of the iodine ion, for which the largest contributions come from the five- and seven-photon curves. Indeed, as can be seen in Figure 3e, the shape of the six-photon curve cannot reproduce the experimental signal."

We also added a new subsection in the theory section of the SI, in which we describe the fitting procedure in detail.

D) Further speculations: It is concluded from comparing the shapes of the experimental signals and the calculated curves that CH_3I^+ is produced from 3-photon absorption, CH_3^+ from 5- and 6-photon absorption, and I^+ from 5- and 7-photon absorption. Considering that low-photon absorption is more likely than high-photon absorption, these conclusions would imply (everything else being equal) that the CH_3I^+ signal should be stronger than the CH_3^+ signal which, in turn, should be stronger than the I^+ signal. This is, in fact, the case for the signals in Fig. 2. Maybe that is worth a comment in the manuscript.

We appreciate the referee's comment. In our experiment, it is evident that among the three different ions examined, the parent ion exhibits the highest yield. Its maximum is nearly two orders of magnitude higher than that of CH_3^+ and three orders of magnitude higher than I^+ . To stress this aspect, we report the relative ion yield in Fig. 2, and in Fig. S3 we show the highest ion counts. As noted by the reviewer, this finding aligns with the assignment we made of the dominant multiphoton transitions for each ion. To further address this point, we have included the following sentences:

Results, page 5 (after Fig. 2): "The parent ion CH_3I^+ is the most abundant ion produced in our experiment, since its maximum yield is almost two orders of magnitude higher than the one of the methyl fragment and three orders of magnitude higher than the one of the iodine fragment."

and

Discussions, page 11:

“Assuming that the absorption probability is higher for a low number of photons, these assignments are consistent with the corresponding ion yields observed in our experiments. CH_3I^+ is the most abundant ion produced because it only requires a three-photon transition. The parent ion is then followed by CH_3^+ which requires five-photon absorption, while I^+ is the least produced ion because it primarily requires the absorption of seven photons.”

4) Editorial comments: The manuscript and especially the SI could benefit from proof reading. There are unexplained abbreviations such as “SH-FROG” in the Experimental setup on page 16 (page 1 of the SI) and the Experimental Setup needs more references. Mismatch between Fig. 2 and Fig. S3 is already mentioned above. The figure references are wrong on page 19 (page 4 of the SI) where S2 should be S3 and S3 should be S4.

We thank the reviewer for the suggestions. "SH-FROG" refers to Second Harmonic Frequency Resolved Optical Gating. In response to the reviewer's feedback, we have updated the manuscript to clarify the acronym and have included appropriate references in the experimental setup description. As mentioned above, we are grateful to the reviewer for identifying the mistake in Figure S3. Furthermore, we thank the reviewer for highlighting the typographical errors in Figures S2, S3, and S4. We have now rectified these errors in the manuscript accordingly.

5) Impact and interest for the general readership:

Iodomethane is a popular prototype molecule, and the work will impact the field of femtochemistry of small gas-phase molecules because of 1) the identification of signatures of the conical intersection between the neutral states because of the due to the field-free propagation at the end of the UV pulse, and 2) the showcasing of an ultrafast stabilization scheme against dissociation by few-femtosecond ionization. For the general readership (within femtochemistry) the short UV pulse will be appreciated, and so would be a scheme for benchmarking of trajectory calculations (to cite the abstract). However, as pointed out above, to achieve this, the trajectory calculations must be described in more detail.

We are very grateful to the reviewer for acknowledging the importance of our work and the potential impact on a general readership. We hope that by having addressed all the raised criticisms and by having substantially improved the description of the trajectory calculations, including an additional modelling of the multiphoton probing step, the manuscript is now suitable for publication in Nature Communications.

Reviewer #3 (Remarks to the Author):

REVIEWER COMMENTS

Reviewer #1 (Remarks to the Author):

I appreciate the response of the authors to my earlier comments. However, given the rather convoluted interpretation of the measurements and the lack of direct comparison between experiment and an explicit calculation of the measurement observable, I don't find that the manuscript provides a sufficiently direct measurement of the dynamics to really gain new insight into the non-adiabatic dynamics. I appreciate the time resolution that the authors achieve and the technical challenges associated with it, but I just don't think that it has led to insight worthy of publication in Nature Communications. The experimental approach is very well established, and the new aspect here is simply the very short pump pulse duration. I still don't find figure 3 very compelling or clear (the agreement between the data and calculations does not provide a clear or compelling picture of the non-adiabatic dynamics that the authors aim to uncover), and in my opinion, this is one of the central results of the manuscript.

So, in conclusion, I still don't find the manuscript suitable for publication in Nature Communications.

Reviewer #2 (Remarks to the Author):

In the revised version of manuscript Few-femtosecond time-resolved study of the UV-induced dissociative dynamics of iodomethane, the authors have made many improvements, and I appreciate the effort and willingness to do so. However, I do think that the presentation should be further improved in several respects before the manuscript can be considered for publication.

1) The trajectory simulations: The new paragraph "Finally, at each time step, we associate a trajectory leading to an ion i with a specific number of absorbed NIR photons n , ... For trajectories with multiple dissociation channels open, we assign it only to the ion fragment with the highest threshold." is certainly a huge help to the reader but further help is needed. I think the authors should appreciate how surprising Fig. 3 may appear to the reader compared to Fig. 2, as it shows the exact opposite trend regarding yields of the fragment I^+ in the asymptotic region and, thus, detail the reason(s). For $n=6$, both I^+ and CH_3^+ ions are created but far more trajectories end up in I^+ than in CH_3^+ in the asymptotic region. Is this because more I^+ states are available? Is it because of the strategy described in the last sentence in the new paragraph cited above? Please state the reason(s) for this outcome of the simulations explicitly and explain why the cited assignment strategy was chosen for trajectories with multiple dissociation channels open.

2) The TDSE simulations: I don't really understand what is going on here. On one hand, it says that the transition amplitudes are represented by Dyson norms but on the other hand, assignment to the fragment channels is done by using Eq. 1. Why is that latter necessary? The authors state that the results confirm the behavior of the main text but that is not quite true for CH_3^+ . In figure S9, bottom panel, the methyl fragment is NOT "shown to slowly rise over time" as stated in the text below the figure. Please describe the figure accurately and comment on the fact that the methyl fragment apparently decays to zero at 30 fs in contraction to both experimental and trajectory results.

3) Figures:

A) I appreciate the new figure S5, but I ask the authors to help the readers by interchanging the colors red and cyan, so they match the colors in Fig. 1, where the neutral potential-energy curves are red for I^* and cyan for I .

B) In many of the theoretical curves, it states that the curves in each panel are normalized to the maximum of one (or something similar). However, this is not the case for: i) The solid curves in Fig. 3 (d)-(f), which leads to the odd situation that the dashed curves in (e) and (f) reach above the solid curves. This mistake arises from ii) the curves in Fig. S6 not being normalized appropriately. iii) The curve in the middle panel in Fig. S9. Please fix this.

4) The new data analysis on Relative contribution of the theoretically extracted probabilities: This is a brute-fitting force of the trajectory curves to the experimental data. I had in my first report envisioned a more intuitive analysis also including flexibility on the time axis (especially for CH₃⁺) but be that as it may. However, the section is full of mistakes: i) Table S2 is referred to in the text as Table R1. ii) For the parent ion, the text says that "only A₃ is significantly different from zero" while Table S2 says that A₃=9.3, A₄=5.5, and A₅=10. iii) For I⁺, the text says that "coefficients A₄, A₅ and A₇ are one order of magnitude higher than A₆", while Table S2 says that A₄=0. iv) For CH₃⁺, the reported values for A₅ and A₆ in Table S2 do not match Fig. S8, bottom panel. In this panel, the long-time yield is ca. 0.07 (pC/shot) above the base line, so the A₅ and A₆ coefficients must have values of this magnitude, since the n=5 and n=6 curves for CH₃⁺ in Fig S6 have maximum values on the order of unity (especially after appropriate normalization of the curves in Fig. S6 as mentioned above). Please fix this.

Manuscript ID: NCOMMS-23-62707

Title: Few-femtosecond time-resolved study of the UV-induced dissociative dynamics of iodomethane

Dear Editor,

Thank you for giving us the opportunity to further improve our manuscript. As requested, we are now sending a reply to the reviewers, in which we specifically address the simulations and the methodology used to sort the trajectories. We strongly believe that this new iteration has significantly improved the clarity of the methods used to analyze the theoretical results and compare them with the experiment, which ultimately improves the overall quality of our manuscript.

We are looking forward to receiving positive feedback from you.

Sincerely,

Francesca Calegari and Fernando Martin (on behalf of all the co-authors)

List of changes:

In reply to the Reviewers' comments, we have revised the manuscript and introduced the following changes.

Main text:

- Figure 3: Normalization of the solid curves in panel (d-f) has been corrected.
- Page 8: We removed the sentence concerning the assignment to different ions.

Supplementary Information:

- Figure S5: The color of the weighted average solid lines has been swapped.
- Figure S6: Each set of curves is normalized to a value of 1.
- Page 26: The fitting procedure has been now carried out on normalized experimental data. The new retrieved values for the coefficients are now reported in Table S2 and the figure has been updated with normalized experimental data. The corresponding text in the SI has been modified and the description of the normalization and fitting procedure has been improved.
- Page 30: We added a paragraph in the TDSE section to clarify the criterion used to assign fragment channels.
- Page 31: A more detailed discussion on the comparison between the retrieved time-dependent probability of producing the CH_3^+ ion and the experimental results is reported.
- Page 32: We added a new paragraph where we compare two different criteria to assign trajectories to specific ions. Figure S9 has been modified accordingly.

Blue: reviewers' comment

Red: change in manuscript

Black: authors' answer

REVIEWER COMMENTS:

Reviewer #1 (Remarks to the Author):

I appreciate the response of the authors to my earlier comments. However, given the rather convoluted interpretation of the measurements and the lack of direct comparison between

experiment and an explicit calculation of the measurement observable, I don't find that the manuscript provides a sufficiently direct measurement of the dynamics to really gain new insight into the non-adiabatic dynamics. I appreciate the time resolution that the authors achieve and the technical challenges associated with it, but I just don't think that it has led to insight worthy of publication in Nature Communications. The experimental approach is very well established, and the new aspect here is simply the very short pump pulse duration. I still don't find figure 3 very compelling or clear (the agreement between the data and calculations does not provide a clear or compelling picture of the non-adiabatic dynamics that the authors aim to uncover), and in my opinion, this is one of the central results of the manuscript.

So, in conclusion, I still don't find the manuscript suitable for publication in Nature Communications.

We thank the reviewer for acknowledging the novelty of using ultrashort UV pulses and the technical challenges associated with performing time-resolved experiments with such extreme time resolution. However, we kindly disagree with the reviewer when they state a lack of new findings. Indeed, this is the first exploration of the ultrafast dissociative dynamics of CH₃I in the complete absence of the pump field. Despite the approximations done in modeling the multiphoton probing step, there are substantial (direct and indirect) information that can be obtained from the comparison between experiment and theory including 1) a key ultrafast stabilization mechanism against dissociation and 2) the time window in which the molecular wavepacket reaches the CI. We have shown that no significant REMPI is expected in the time window of interest, and therefore we believe that our conclusions are valid. A more advanced theoretical modeling would certainly provide a more "compelling" story, however the theoretical calculations performed here are state of the art. Calculating the actual IR-induced multiphoton ionization step as a function of the pump-probe delay and the subsequent fragmentation dynamics at each delay for such a complex molecule is completely out of reach with the current theoretical and computational capabilities. To our knowledge, such a complete study has only been reported for the H₂ molecule, where a full quantum mechanical calculation is possible. We would like to point out again that despite the complexity of the experiment and the simplicity of our theoretical modeling, we are able to advance our knowledge of the UV induced dissociative dynamics of CH₃I, and we can demonstrate a scheme to stabilize the molecule against the otherwise inevitable UV induced fragmentation.

Reviewer #2 (Remarks to the Author):

In the revised version of manuscript Few-femtosecond time-resolved study of the UV-induced dissociative dynamics of iodomethane, the authors have made many improvements, and I appreciate the effort and willingness to do so. However, I do think that the presentation should be further improved in several respects before the manuscript can be considered for publication.

We thank the reviewer for this comment.

1) The trajectory simulations: The new paragraph "Finally, at each time step, we associate a trajectory leading to an ion i with a specific number of absorbed NIR photons n , ... For trajectories with multiple dissociation channels open, we assign it only to the ion fragment with the highest threshold." is certainly a huge help to the reader but further help is needed.

We are grateful for the reviewer's comments and for their careful review of our manuscript, which helped us again in substantially improving our work.

I think the authors should appreciate how surprising Fig. 3 may appear to the reader compared to Fig. 2, as it shows the exact opposite trend regarding yields of the fragment I⁺ in the asymptotic region and, thus, detail the reason(s). For $n=6$, both I⁺ and CH₃⁺ ions are created but far more trajectories end up in I⁺ than in CH₃⁺ in the asymptotic region. Is this because more I⁺ states are available? Is it because of the strategy described in the last sentence in the new paragraph cited

above? Please state the reason(s) for this outcome of the simulations explicitly and explain why the cited assignment strategy was chosen for trajectories with multiple dissociation channels open.

First of all, we deeply apologize with the reviewer because the sentence “For trajectories with multiple dissociation channels open, we assign it only to the ion fragment with the highest threshold” should not have been written in the main text, but in the SI. Indeed, this sentence was only referring to the TDSE calculations, which we reported at the end of the previous version of the SI. In the simple model described in the main text, we correctly assigned the trajectories to I^+ or CH_3^+ by identifying the potential energy surface that is compatible with the energy acquired by the nuclei. Only in the TDSE calculations we applied a coarser procedure and assigned the trajectories to the ion fragment with the highest threshold value. This was done for computational reasons, as the TDSE calculations are far more expensive than the model calculations described in the main text. For this reason, we have now removed the above sentence from the main text.

We agree with the reviewer that it is somewhat surprising to find more trajectories leading to I^+ than to CH_3^+ in the asymptotic region. However, this is consistently found from the simple model presented in Fig. 3 and the more elaborate TDSE calculations. In short, the answer to the reviewer’s question is that the reason for this behavior is that, in both kinds of simulations, the number of states dissociating into I^+ is considerably larger than the number of states dissociating into CH_3^+ .

In any case, to discard that the way we assign the trajectories to the production of I^+ and CH_3^+ in the TDSE calculations could lead to an accidental agreement between the results of the latter and those of the simple model presented in the main text, we have repeated the TDSE calculation by using **exactly the same** assignment criterion as in the simple model. In other words, instead of assigning the trajectories only to the ion fragment with the highest threshold value, we have assigned the trajectories to I^+ or CH_3^+ by identifying the potential energy surface that is compatible with the energy acquired by the nuclei. The results are shown in figure R1, which also includes those presented in the previous version of the manuscript. By using the new more selective criterion to assign the trajectories, the absolute value for the production of I^+ decreases and that of CH_3^+ increases (not seen because of normalization), because some of the trajectories that were initially assigned to I^+ are now more correctly assigned to CH_3^+ . However, the qualitative behavior is very similar. In particular, the I^+ yield still dominates over the CH_3^+ one in the asymptotic region. So, we can safely discard that the way the assignment of the dissociation channels is made is the reason for the observed behavior.

One may also wonder if the fact that I^+ holds a spherical potential, imposing restrictions to the values of the angular momentum that the ionized electron can have, in contrast with CH_3^+ that holds a multi-centric potential imposing no restrictions, can make a significant difference. In both the simple model and the more elaborate TDSE calculations this effect is not considered because none of these methods includes the electronic continuum of the system explicitly. However, it is very unlikely that this is the reason for the observed behavior. Indeed, considering the most typical situation in which the electron has an initial angular momentum of 1 in neutral iodine (corresponding to an electron in the 5p orbital) and 6 photons are absorbed, at least all angular momenta from $l=0$ up to $l=7$ should be accessible for the ionized electron. For CH_3^+ , formally there is not such a restriction, but it would be very unlikely that electrons with angular momentum larger than 7 are created after absorption of 6 photons from the 2p electron of the C atom in the CH_3 fragment.

To clarify this point, we have included the following paragraph in the SI:

Both the TDSE simulations and the simpler model described in the main text predict that there are more trajectories leading to I^+ than to CH_3^+ at long times, in contrast to the experimental finding. The reason for this behavior is that in both types of simulations the number of states dissociating into I^+ is considerably larger than the number of states dissociating into CH_3^+ .

We have also replaced Fig. S9 with a new one showing the results of the TDSE calculations in which the assignment of the different dissociation channels has been done (as in the simple model reported in the main text) by identifying the correct potential energy surface compatible with the energy acquired by the nuclei.

Figure R1: Comparison of the TDSE simulation results by using different criteria to assign trajectories to the different ions. The solid lines result from the old criterion (referred to in the caption as Highest Threshold), which selects trajectories with a total energy higher than the dissociation threshold. Dashed lines result from the new criterion (referred to as PES in the caption), also used for the simpler model, in which we identify the correct potential energy surface compatible with the energy acquired by the nuclei. Each curve probability has been normalized to a maximum of 1 in order to better compare the results of the two selection criteria.

2)The TDSE simulations: I don't really understand what is going on here. On one hand, it says that the transition amplitudes are represented by Dyson norms but on the other hand, assignment to the fragment channels is done by using Eq. 1. Why is that latter necessary?

We first note that our approximate way to describe multiphoton ionization by the probe pulse assumes that the nuclei move classically after excitation of the system by the pump pulse. In other words, our solution of the TDSE only concerns the electrons, not electrons and nuclei as it would be the case in a full quantum mechanical description, which is not possible for a molecule of this size. As a consequence, the “electronic” TDSE must be solved for each nuclear geometry reached in a given trajectory (150) and this for all trajectories (1,000), which amounts to 150,000 TDSE calculations. This provides the multiphoton ionization probabilities to the different states of the cation for all considered nuclear geometries. However, these probabilities do not tell us if the molecule will dissociate or not. To know this, one has to consider that, at the instant of ionization by the probe pulse, the nuclei have already acquired a certain amount of kinetic energy E_{kin} . This quantity is easily extracted from the calculated nuclear trajectories. If the sum $E_{kin} + E_{pot}$ is larger than the dissociation energy D_i in a given state of the cation (eq. 1 of the main text), multiphoton ionization will be followed by dissociation leading to fragment i . If not, the molecular cation will not dissociate.

We apologize for not having explained this in more detail in the revised version. We have now included the following paragraph in the SI (section “Time-dependent Schrödinger equation simulations”) to make this point clearer:

It is worth mentioning that in this description of multiphoton ionization by the probe pulse, the electronic TDSE must be solved for each nuclear geometry reached in a given trajectory (150) and for all the trajectories (1,000), which amounts to 150,000 TDSE calculations. This provides the multiphoton ionization probabilities to the different states of the cation for all considered nuclear geometries. However, these probabilities do not tell us if the molecule will subsequently dissociate or not. To know this, one has to consider that, at the instant of ionization by the probe pulse, the nuclei have already acquired a certain amount of kinetic energy E_{kin} . This quantity is easily extracted from the calculated nuclear trajectories. When the sum $E_{kin} + E_{pot}$ is larger than the dissociation energy D_i in a given state of the cation (eq. 1 of the main text), multiphoton ionization will be followed by dissociation leading to fragment i . Otherwise, the molecular cation will not dissociate.

The authors state that the results confirm the behavior of the main text but that is not quite true for CH₃⁺. In figure S9, bottom panel, the methyl fragment is NOT “shown to slowly rise over time” as stated in the text below the figure. Please describe the figure accurately and comment on the fact that the methyl fragment apparently decays to zero at 30 fs in contraction to both experimental and trajectory results.

We agree with the reviewer that for the methyl ion at relatively large delays (above 20-30 fs), the new calculations deviate both with what observed experimentally and with what has been obtained with the previous model. However, we don't extract any crucial information at these large delays and therefore we expect our interpretation to remain unaffected by this discrepancy. We have now modified the corresponding paragraph in the supplement as follows:

Additionally, the methyl fragment is shown to rise during the first 15 fs and decay to zero at around 30 fs. The rising of the signal agrees with the behavior reported in Fig. 2 for the four-photon and the five-photon contribution, while the former six-photon contribution seems to be missing. This fast decay does not agree with what is observed experimentally, however we can safely assume that our interpretation of the mechanisms occurring at shorter time delays (NIR-induced stabilization and passage through the CI), where the agreement with theory is satisfactory, remains unaffected. As reported in the main manuscript for the simpler model, the peak in the experimental signal for the methyl ion at zero-time delay is due to an internal conversion mechanism in the molecular cation, which is not included in the TDSE calculation and therefore is missing in the curve reported in the bottom panel of Fig. S9.

3) Figures:

A) I appreciate the new figure S5, but I ask the authors to help the readers by interchanging the colors red and cyan, so they match the colors in Fig. 1, where the neutral potential-energy curves are red for I* and cyan for I.

We have changed the colors of the curves as requested.

B) In many of the theoretical curves, it states that the curves in each panel are normalized to the maximum of one (or something similar). However, this is not the case for: i) The solid curves in Fig. 3 (d)-(f), which leads to the odd situation that the dashed curves in (e) and (f) reach above the solid curves. This mistake arises from ii) the curves in Fig. S6 not being normalized appropriately. iii) The curve in the middle panel in Fig. S9. Please fix this.

We apologize for the mistakes, obviously something went wrong when reporting the normalized curves. We are very grateful to the referee for identifying these inconsistencies. We have now carefully revised the normalization of the theoretical curves and we have updated the corresponding figures.

4) The new data analysis on Relative contribution of the theoretically extracted probabilities: This is a brute-fitting force of the trajectory curves to the experimental data. I had in my first report envisioned a more intuitive analysis also including flexibility on the time axis (especially for CH₃⁺) but be that as it may. However, the section is full of mistakes: i) Table S2 is referred to in the text as Table R1. ii) For the parent ion, the text says that “only A₃ is significantly different from zero” while Table S2 says that A₃=9.3, A₄=5.5, and A₅=10. iii) For I⁺, the text says that “coefficients A₄, A₅ and A₇ are one order of magnitude higher than A₆”, while Table S2 says that A₄=0. iv) For CH₃⁺, the reported values for A₅ and A₆ in Table S2 do not match Fig. S8, bottom panel. In this panel, the long-time yield is ca. 0.07 (pC/shot) above the base line, so the A₅ and A₆ coefficients must have values of this magnitude, since the n=5 and n=6 curves for CH₃⁺ in Fig S6 have maximum values on the order of unity (especially after appropriate normalization of the curves in Fig. S6 as mentioned above). Please fix this.

We sincerely thank the reviewer for their invaluable feedback. The discrepancies between the text and the coefficient values reported in the table are due to the use of normalized curves to fit experimental quantities with physical dimensions. This normalization sometimes results in the fit procedure enhancing curves with very low relative amplitude. For instance, the five-photon curve for CH₃I⁺ is extremely low compared to the three- and four-photon curves in Figure S6, leading the fit routine to select an unreasonably large coefficient for the five-photon curve compared to the other two. A similar issue occurs for CH₃⁺, where the experimental data shows very small absolute quantities, resulting in extremely small fitting coefficients. We apologize for the lack of clarity on this point.

To provide more intuitive coefficient values, we have normalized the experimental traces to their maximum value and fitted them with a linear combination of the normalized theory curves reported in Figure S6. This approach makes the comparison with theoretical curves more consistent. Additionally, we removed the constant baseline by subtracting the average yield extracted at negative delays for each ion, thus eliminating the need for the A₀ coefficient.

While the fit procedure remains essentially unchanged, the new coefficients more intuitively reflect the weight of the various contributions. The revised coefficients, along with their standard errors, are reported in the updated Table S2.

Parameters	CH₃I⁺	I⁺	CH₃⁺
A ₃	0.80 ± 0.01	0	0
A ₄	0.10 ± 0.01	0	4.0e-10 ± 0.5
A ₅	0.1 ± 0.5	2.8 ± 0.6	0.35 ± 0.03
A ₆	0	4.02e-09 ± 0.1	0.29 ± 0.01
A ₇	0	0.62 ± 0.1	0.1352 ± 0.18
A ₈	0	1.9e-11 ± 0.3	0

Table S2: Coefficients obtained by fitting the set of normalized curves from figure S6 to the experimental data normalized to the maximum for each ion. A_n represents the relative contribution of the n-photon curves from Figure S6, along with the standard error obtained by the fitting procedure.

By normalizing the ion yields, for CH₃I⁺ we observe a clear major contribution from the three-photon channel and a minor one from the four-photon channel. The five-photon contribution has a very large uncertainty. For I⁺, the results align with what has been reported before: the five-photon and seven-

photon contributions dominate. Finally, for CH_3^+ , we retrieve major contributions from the five- and six-photon channels.

The corresponding supplementary information section has been updated, the description of both the normalization and the fitting procedure has been improved, and Table S2 and Figure S8 have been modified accordingly. Additionally, the A_4 coefficient has been correctly reported in the text.

We sincerely hope that we have now properly addressed all the reviewers' concerns and that the manuscript in its present version is suitable for publication in Nature Communications.

REVIEWERS' COMMENTS

Reviewer #2 (Remarks to the Author):

In the second revised version of manuscript Few-femtosecond time-resolved study of the UV-induced dissociative dynamics of iodomethane, the authors have made many improvements, and once again I appreciate the effort and willingness to do so. I think that the manuscript is now in very good shape, and I recommend publication after consideration of a few minor points. I agree with the Reviewer #1 that perhaps another journal could be more appropriate but, on the hand, the manuscript combines very high-level experiments with a variety of theoretical approaches for interpreting the experimental results. Short of a full simulation of the experiment, which is currently not possible, I believe that the kind of 'patchwork' analysis presented in the manuscript is exemplary and therefore suitable for publication in Nature Communications.

Minor points:

1) The new statement "the number of states dissociating into I+ is considerably larger than the number of states dissociating into CH3+" at the very end of the SI is certainly very helpful. I think it would really increase the readability of the manuscript if this statement was also given in the main manuscript in connection with either Fig. 1 or Fig. 3, so the reader does not (as I did) get caught up in trying to interpret Fig. 3 without this knowledge.

2) The revised Table S2 is much more intuitive than the previous version, thank you. I have a few comments: a) It would increase the readability if the very small numbers in the 1e-9 to 1e-11 range in A6 and A8 for I+ and in A4 for CH3+ were replaced by zeros. b) For these three A coefficients as well as A5 for CH3I+ and A7 for CH3+, the lower bounds given would result in negative A coefficients. This is clearly unphysical, and I suggest reporting the uncertainty in a different manner (of if the A coefficients were not kept positive in the fit procedure, to comment on this). c) There is a misprint in the A5 coefficient for I+; it should be ca. ten times smaller.

Manuscript ID: NCOMMS-23-62707

Title: Few-femtosecond time-resolved study of the UV-induced dissociative dynamics of iodomethane

Dear Editor,

Thank you for the positive feedback regarding our manuscript. Here we address the remaining comments from reviewers.

Sincerely,
Lorenzo Colaizzi (on behalf of all the co-authors)

Blue: reviewers' comment

Red: change in manuscript

Black: authors' answer

REVIEWER COMMENTS:

Reviewer #2 (Remarks to the Author):

In the second revised version of manuscript Few-femtosecond time-resolved study of the UV-induced dissociative dynamics of iodomethane, the authors have made many improvements, and once again I appreciate the effort and willingness to do so. I think that the manuscript is now in very good shape, and I recommend publication after consideration of a few minor points. I agree with the Reviewer #1 that perhaps another journal could be more appropriate but, on the hand, the manuscript combines very high-level experiments with a variety of theoretical approaches for interpreting the experimental results. Short of a full simulation of the experiment, which is currently not possible, I believe that the kind of 'patchwork' analysis presented in the manuscript is exemplary and therefore suitable for publication in Nature Communications.

We are grateful for your recognition of the high-level experiments and the comprehensive theoretical approaches we employed to interpret the results.

We are pleased to hear that you find the manuscript in very good shape and suitable for publication in Nature Communications. Your insightful comments have significantly contributed to enhancing the quality of our work. We have addressed the minor points you mentioned and believe that these revisions further strengthen our manuscript.

Minor points:

1) The new statement "the number of states dissociating into I⁺ is considerably larger than the number of states dissociating into CH₃⁺" at the very end of the SI is certainly very helpful. I think it would really increase the readability of the manuscript if this statement was also given in the main manuscript in connection with either Fig. 1 or Fig. 3, so the reader does not (as I did) get caught up in trying to interpret Fig. 3 without this knowledge.

We thank the reviewer for their insight. We have added this statement in the first sentence of the second paragraph from the end of the Results section on Page 6.

"Using this simple approach, we predict that there are many more trajectories leading to I⁺ than to the other ions."

2) The revised Table S2 is much more intuitive than the previous version, thank you. I have a few comments: a) It would increase the readability if the very small numbers in the $1e-9$ to $1e-11$ range in A_6 and A_8 for I^+ and in A_4 for CH_3^+ were replaced by zeros. b) For these three A coefficients as well as A_5 for CH_3I^+ and A_7 for CH_3^+ , the lower bounds given would result in negative A coefficients. This is clearly unphysical, and I suggest reporting the uncertainty in a different manner (of if the A coefficients were not kept positive in the fit procedure, to comment on this). c) There is a misprint in the A_5 coefficient for I^+ ; it should be ca. ten times smaller.

Thank you for your valuable suggestions regarding Supplementary Table 2. We have carefully considered your comments and made the following revisions and clarifications:

- a) We have replaced the very small numbers with zeros to enhance readability, as you suggested.
- b) You are correct in noting that the A coefficients were kept positive during the fitting procedure. The standard nonlinear optimization routines that we are aware of report symmetric confidence intervals based on the Jacobian near the found optimum, regardless of whether those intervals extend beyond the parameter bounds. In our case, a curve whose coefficient has a fit uncertainty interval including zero does not contribute significantly to the total fit.
- c) We have re-examined Supplementary Table 2 and realized that the previously shown coefficients were mistakenly the result of a fit using an unnormalized basis set. In the previous version, the obtained values were taking into account the amplitude of its corresponding curve in Supplementary Fig. 6. (that is why A_5 was surprisingly large, because its basis set curve had a low amplitude.) Now we use the normalized basis set as it is described in the “Relative contribution of the theoretically extracted probabilities” section of Supplementary Note 5 and provide the correct values in the table, which simplifies the interpretation. The correction is notable only for the ratio between A_5 and A_7 for I^+ , while the amplitudes of the relevant basis curves for the other ions were already close to 1. We would like to stress that the change only affects the coefficients (Supplementary Table 2), while the fitted curves and Supplementary Figure 8 remain unaffected.